# ON-DEVICE COLLABORATIVE LANGUAGE MODELING VIA A MIXTURE OF GENERALISTS AND SPECIALISTS

## ABSTRACT

On-device LLMs have gained increasing attention for their ability to enhance privacy and provide a personalized user experience. To facilitate private learning with scarce data, Federated Learning has become a standard approach. However, it faces challenges such as computational resource heterogeneity and data heterogeneity among end users. We propose CoMiGS (**Co**llaborative learning with a **Mi**xture of **G**eneralists and **S**pecialists), the first approach to address both challenges. A key innovation of our method is the bi-level optimization formulation of the Mixture-of-Experts learning objective, where the router is optimized using a separate validation set to ensure alignment with the target distribution. We solve our objective with alternating minimization, for which we provide a theoretical analysis. Our method shares generalist experts across users while localizing a varying number of specialist experts, thereby adapting to users' computational resources and preserving privacy. Through extensive experiments, we show CoMiGS effectively balances general and personalized knowledge for each token generation. We demonstrate that CoMiGS remains robust against overfitting—due to the generalists' regularizing effect—while adapting to local data through specialist expertise. We open source our codebase for collaborative LLMs.

## 1 INTRODUCTION

Large Language Models (LLMs) have been showing great success serving as foundation models, evidenced by their capability to understand a wide range of tasks, such as ChatGPT (OpenAI, 2023), Claude (Anthropic, 2023), Gemini (DeepMind, 2023) and etc. However, cloud-based inference introduces significant delays for end users, and it often fails to meet their personalized needs (Ding et al., 2024; Iyengar & Adusumilli, 2024). Recently, there has been growing interest in deploying LLMs on edge devices, which offer benefits like lower latency, data localization, and more personalized user experiences (Xu et al., 2024). For instance, Apple (2024) recently launched on-device foundation models as part of its personal intelligence system. Meta (2024), Qwen (2024) newly released lightweight models with less than 3B parameters targeting edge AI.

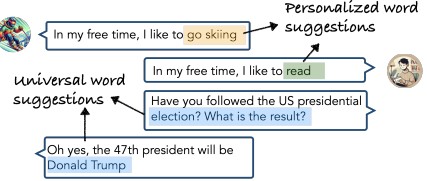

Figure 1: Chat box between two users with different characteristics. Next word prediction for smart keyboards should be tailored to users' topic preferences for personalization. However, to ensure factual accuracy and linguistic consistency, the results of next word prediction should maintain universality.

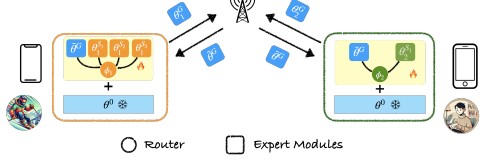

Figure 2: Diagram of our proposed method CoMiGS illustrated with a simplified 2-heterogenous-models setup (corresponding to the two users in Fig. 1). Generalist experts ($\theta_1^G$, $\theta_2^G$) are aggregated across users, and specialist experts ($\{\theta_1^{S_i}\}_{i=1}^3$, $\{\theta_2^{S_1}\}$) and Routers ($\phi_1$, $\phi_2$) are kept local.

On-device LLMs present challenges such as limited and variable computational resources, scarce and heterogeneous local data, and privacy concerns related to data sharing (Peng et al., 2024; Wagner et al., 2024). Fine-tuning is typically performed on-device to quickly adapt to users' individual needs. While data sharing is a common solution to address local data scarcity, on-device data is often privacy-sensitive and must remain on the device. To overcome this, Federated Learning has been proposed as a method for enabling collaborative learning while preserving user privacy, allowing end users to collaborate by sharing model parameters (Chen et al., 2023; Zhang et al., 2023).

Collaboration between end devices introduces challenges like model (Cho et al., 2023; Bai et al., 2024) and data heterogeneity Wagner et al. (2024). Moreover, in language modeling, decisions on collaboration vs. personalization occur at the word level. For instance, as shown in Fig. 1, the prompt "In my free time, I like to" should yield user-specific predictions, while factual statements, such as the U.S. presidential election result, should remain universal.

Towards addressing these challenges, we propose a novel **Co**llaborative learning approach via a **Mi**xture of **G**eneralists and **S**pecialists (CoMiGS). Our approach utilizes a Mixture-of-Experts architecture and allows users to share some expert modules while keeping other modules user-specific, thus providing personalized solutions. We name the shared part *generalists* and the user-specific part *specialists*. Like all previous works, the generalists and specialists are simply LoRA modules (Hu et al., 2021). At the same time, as long as the shared part can be aggregated, the user-specific part can be of different sizes, which can be adapted to various device capacities, as illustrated by different numbers of specialists across users in Fig. 2.

We further notice a hierarchical structure between the router and the experts: the router dynamically assigns tokens based on emerging expert specializations, while the experts refine their roles to optimize token processing under the router's guidance. Towards addressing this, we formulate our learning objective as a bi-level optimization problem and propose a new first-order algorithm based on alternating minimization as a solution. Our method enjoys convergence guarantees and is resource-efficient for deployment.

**Contributions:**

- We propose a novel approach (CoMiGS) for on-device personalized collaborative fine-tuning of LLMs. Key parts of our approach are: 1) an innovative bi-level formulation of the MoE learning objective (Section 2.1); 2) a new algorithm based on alternating minimization (Alg.1); 3) a theoretical analysis with a proof showing linear convergence rate under suitable assumptions (Section 2.3).

- Our collaborative framework effectively addresses both *data heterogeneity* (Section 3.1), concerning diverse local data distributions across users, and *computational resource heterogeneity* (Section 3.2), with respect to varying local model architectures, making it the first model to accomplish both.

- Our framework separates model heterogeneity from data quantity (Section 3.3). Users with larger local datasets benefit from a bigger model, while users with more powerful models but smaller datasets are less prone to overfitting.

- CoMiGS is resource-efficient: it adds marginal ($+1.25\%$) computational overhead and memory requirement compared to FedAvg, while reducing communication costs by 50% (Appendix C).

## 2 METHOD

Building on the hierarchical insights of MoE learning, we formulate our learning objective into a bi-level optimization problem, where expert parameters and routing parameters are updated using training sets and validation sets respectively. We further let experts diversify into generalists and specialists via parameter aggregation or localization. As the problem solver, we provide a multi-round gradient-based algorithm, of which the pseudo codes are presented in Appendix A.

### 2.1 A BI-LEVEL FORMULATION

Instead of learning routing and expert parameters simultaneously like the conventional way in LLMs (Zoph et al., 2022; Fedus et al., 2022), we update the two sets of parameters in an alternating fashion. We observe *a natural hierarchy between the experts and the router*: the assignment of tokens to experts depends on the router's outputs, while the experts' parameters are updated based on the assigned tokens. In this way, the experts' development follows the router's decisions, establishing an

inherent leader-follower structure. Following Von Stackelberg (2010), we formulate the hierarchical problem as a bi-level optimization objective as follows:

$$\min_{\Phi} \sum_i \mathcal{L}(\boldsymbol{X}_i^{\text{valid}}, \boldsymbol{\Theta}^{\star}(\boldsymbol{\Phi}), \boldsymbol{\phi}_i) \qquad \text{(upper)}$$

$$s.t. \ \boldsymbol{\Theta}^{\star}(\boldsymbol{\Phi}) \in \arg\min_{\boldsymbol{\Theta}} \sum_i \mathcal{L}(\boldsymbol{X}_i^{\text{train}}, \boldsymbol{\theta}^G, \boldsymbol{\theta}_i^S, \boldsymbol{\phi}_i) \qquad \text{(lower)}$$

where $\mathcal{L}$ is the language modeling loss. $\boldsymbol{X}_i^{\text{train}}$ and $\boldsymbol{X}_i^{\text{valid}}$ are local training and validation sets respectively. The routing parameters $\boldsymbol{\Phi} = \{\boldsymbol{\phi}_i\}$ are updated based on the validation loss, which reflects the target distribution (upper optimization), while the expert parameters $\boldsymbol{\Theta} = \boldsymbol{\theta}^G \cup \{\boldsymbol{\theta}_i^S\}$ are updated using the training loss (lower optimization). This formulation further brings in the following benefits: 1) routing parameters can be updated less frequently and thus less prone to overfit; 2) it handles situations where target distributions differ from training distributions

## 2.2 OUR ALGORITHM

To solve our bi-level problem, we use alternating updates of the two sets of parameters. The pseudo-code of our proposed algorithm is detailed in Alg.1 in the Appendix.

**Alternating Update of $\boldsymbol{\Theta}$ and $\boldsymbol{\Phi}$.** Alternating update of two sets of parameters is a standard way to solve bi-level optimization problems (Chen et al., 2021). The alternating updates of expert and routing parameters are performed using local training and validation sets separately. To simplify notations, we denote $f_{\text{valid}}(\boldsymbol{\Theta}, \boldsymbol{\Phi}) := \sum_i \mathcal{L}(\boldsymbol{X}_i^{\text{valid}}, \boldsymbol{\theta}^G, \boldsymbol{\theta}_i^S, \boldsymbol{\phi}_i)$ and $f_{\text{train}}(\boldsymbol{\Theta}, \boldsymbol{\Phi}) := \sum_i \mathcal{L}(\boldsymbol{X}_i^{\text{train}}, \boldsymbol{\theta}^G, \boldsymbol{\theta}_i^S, \boldsymbol{\phi}_i)$. Note that in contrast to (upper) bi-level formulation, we allow parameter $\boldsymbol{\Theta}$ to be free in $f_{\text{valid}}$, which makes it easier to optimize. We can write the alternating update steps as follows.

$$\boldsymbol{\Phi}_{k+1} = \arg\min_{\boldsymbol{\Phi}} f_{\text{valid}}(\boldsymbol{\Theta}_k, \boldsymbol{\Phi}), \qquad \boldsymbol{\Theta}_{k+1} = \arg\min_{\boldsymbol{\Theta}} f_{\text{train}}(\boldsymbol{\Theta}, \boldsymbol{\Phi}_{k+1}). \qquad (1)$$

Given that the data is distributed among clients, when optimizing $\boldsymbol{\Theta}_{k+1}$, we first obtain the solutions $\boldsymbol{\theta}_i^G$ and $\boldsymbol{\theta}_i^S$ to local problems, for each client $i$. A parameter aggregation is then performed on the user-specific $\boldsymbol{\theta}_i^G$ via a trusted server to establish a shared $\boldsymbol{\theta}_G$ across all users.

$$\left\{ \tilde{\boldsymbol{\theta}}_i^{G,k+1}, \tilde{\boldsymbol{\theta}}_i^{S,k+1} \right\}_{i=1}^N = \arg\min_{\boldsymbol{\Theta}} f_{\text{train}}(\boldsymbol{\Theta}, \boldsymbol{\Phi}_{k+1}),$$

$$\boldsymbol{\Theta}^{k+1} = \left( \frac{1}{N} \sum_i \tilde{\boldsymbol{\theta}}_i^{G,k+1}, \{\tilde{\boldsymbol{\theta}}_i^{S,k+1}\} \right). \qquad (2)$$

In the next round, each user replaces their $\boldsymbol{\theta}_i^G$ with the global $\boldsymbol{\theta}_G$, while their $\boldsymbol{\theta}_i^S$ remains local.

## 2.3 CONVERGENCE RESULTS

First, we establish a linear rate of convergence under general assumptions on our objectives, that always hold *locally*, when the parameters are close to the training solution (assuming the pretrained model is not far from the fine-tuned models). Then, we show that in the case of *linear experts*, the same optimization procedure possesses *global* linear convergence. The technical details are provided in Appendix G.

**Theorem 2.1** (Convergence under Contraction). *If Assumptions 1, 2 hold, and $\lambda_1 \cdot \lambda_2 < 1$, then the weights $(\boldsymbol{\Theta}_k, \boldsymbol{\Phi}_k)$ generated by alternating updates (1) converge to $(\boldsymbol{\Theta}^{\star}, \boldsymbol{\Phi}^{\star})$ with a linear rate.*

**Theorem 2.2** (Global Convergence for Linear Experts). *If $f_{valid} = f_{train}$ and all the expert modules are linear models, we have a global linear convergence rate for a practical instance of our method.*

## 3 EXPERIMENTS

The experimental setup and relevant details are provided in the Appendix B. Additionlly, we provide an anlysis of the computational and communication Overhead in Appendix C.

## 3.1 Data-Driven Selection: Generalist vs. Specialist

We start by equipping users with the same model architecture locally (GPT-124M or LLama3.2-1B with the same number of LoRA modules), to illustrate the effectiveness of our hierarchical learning of routing and expert parameters. We compare our one generalist one specialist (`CoMiGS-1G1S`) method to the following baselines. In order to match the trainable parameter count of our method, we use 2 times LoRA modules within each user.

- *Upper and lower bounds*: `Pretrained`, `Centralized`
- *Baselines*: `Local`, `FedAvg`, `PCL`, `pFedMoE`, `FDLoRA`
- *Ablations*: `CoMiGS-2S`, `CoMiGS-2G`

### 3.1.1 Result Analysis

The comparison between our method and the baseline methods for models trained on *Multilingual Wikipedia* Wikimedia-Foundation, *SlimPajama* Soboleva et al. (2023), *AG News* (Zhang et al., 2016) or *Common Corpus* (pleias, 2024), including Harvard US Patent dataset (Suzgun et al., 2022) is summarized in Table 2.

**Effectiveness of Our Routing Mechanism.** Depending on the dataset, either `CoMiGS-2G` or `CoMiGS-2S` achieves the best performance. The key advantage over `Local` and `FedAvg` is the layer-wise token-level router, which learns to combine generalists and specialists effectively. This highlights that *how* knowledge is combined is crucial. Although `pFedMoE` also has a learned router, it underperforms even in-distribution because its routing parameters are updated alongside expert parameters, limiting adaptation to the target distribution. When a validation set is unavailable, `CoMiGS` can instead sample new training batches to update routers, maintaining competitive in-distribution performance (see Table 5).

**Token-level Collaborative Decisions Outperform Client-Level.** Compared to the state-of-the-art baseline `PCL` and `FDLoRA`, our method demonstrates a clear performance improvement. While both methods require a separate validation set as in our method to determine collaboration weights, `PCL` determines the weights to combine each client's models iteratively while `FDLoRA` determines the weights for the global and local model at the end of training. Our method, in contrast, decides the collaboration pattern based on each input token, allowing the router weights to co-adapt with the expert parameters throughout training. This enables a more flexible and fine-grained collaboration.

**The Necessity of the Co-existence of Generalists and Specialists.** The performances of `CoMiGS-2G` and `CoMiGS-2S` are not consistent across the different scenarios, while our `CoMiGS-1G1S` can always closely track the best-performing model, which is clearly visualized in Fig. 7. Even for in-distribution tasks, it is unclear whether `CoMiGS-2G` or `CoMiGS-2S` will outperform, suggesting both generalists and specialists are necessary as it is impossible to determine the language structure in advance. Even drastically different users still share many of the same tokens. A data-dependent combination of generalists and specialists is required.

### 3.1.2 Routing Analysis

**Token-wise Analysis.** Fig. 3 visualizes token-level routing for models fine-tuned on SlimPajama. In the *first layer*, function words (e.g., "and", "a", "on", "the") are mostly routed to generalists, while in the *last layer*, content words are more frequently assigned to generalists. This pattern is especially clear for the first two users, trained on math and programming texts, where domain-specific terms are routed to specialists. These results indicate that later-layer experts develop distinct specializations. Notably, only the top choice is shown, so the presence of blue does not mean generalists are unused. Compared to `CoMiGS-2S`, our `CoMiGS-1G1S` produces more consistent results. Additional token-wise routing visualizations, including out-of-distribution tasks, are in Appendix F, with experiments shown in Fig. 12-17.

**Layer-wise Analysis.** Fig. 4 shows the evolution of layer-wise router outputs for generalist and specialist experts in an *out-of-distribution* task, comparing `CoMiGS-1G1S` and `pFedMoE`. As training progresses, `CoMiGS-1G1S` undergoes a *phase transition*: routers initially favor generalists but gradually shift to specialists, a pattern absent in `pFedMoE`, underscoring the importance of our routing mechanism. Different layers converge to distinct expert distributions. With `CoMiGS-1G1S`, some layers consistently favor generalists, reflecting the fact that the target distribution is a union of local training distributions. For *in-distribution* tasks (Fig. 8), early in training, some layers prefer generalists, but near convergence, specialists dominate. This occurs because generalists, trained on

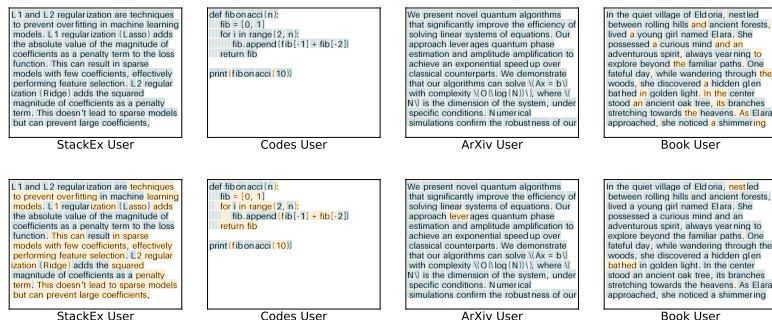

Figure 3: Visualization of in-distribution token-level routing results for `CoMiGS-1G1S` trained on SlimPajama. Tokens are colored with the Top1 expert choice at the first layer (top) and last layer (bottom). Orange denotes the generalist and blue denotes the specialist. Texts are generated by ChatGPT. Further colored text plots are provided in Appendix F.

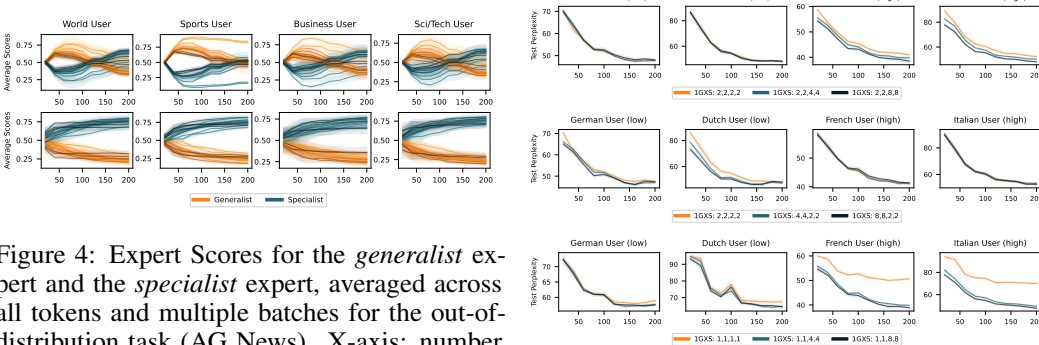

Figure 4: Expert Scores for the *generalist* expert and the *specialist* expert, averaged across all tokens and multiple batches for the out-of-distribution task (AG News). X-axis: number of iterations. Top: `CoMiGS-1G1S`, Bottom: `pFedMoE`. Darker colors indicate deeper layers.

Figure 5: Test Perplexity vs. the number of iterations. Low and high denote data quantity. Legend denotes $n_i$.

more tokens, become knowledgeable sooner, whereas specialists take longer to refine their expertise with limited local data.

## 3.2 ADAPTATION TO COMPUTATIONAL RESOURCE HETEROGENEITY

### 3.2.1 BASELINE COMPARISON

In this section, our focus is to deal with computational resource heterogeneity, where users can have different numbers of experts $n_i$. We denote different experimental setup by specifying the list of $n_i$s. We still keep one generalist expert per device, but the number of specialists can vary across the users (the variation is called One-Generalist-X-Specialists, in short, `CoMiGS-1GXS`). Importantly, the richness of computational resources doesn't always correlate with the complexity of local data. For instance, some users may have ample computational resources but local data in small quantities. In such cases, a crucial objective is to prevent overfitting due to redundant model-fitting abilities.

We compare our approach to two state-of-the-art baselines: `HetLoRA` from Cho et al. (2023) and `FlexLoRA` from Bai et al. (2024), both of which adapt LoRA ranks based on the resource capacity of each user. We compare our method to these baselines by matching the number of tunable parameters, measured as both active and full parameters. For example, to match the full parameter count of `CoMiGS-1GXS` with $(4, 2, 2, 2)$ LoRA experts (rank 8), LoRA modules of ranks $(32, 16, 16, 16)$ would be required. With Top2 routing, to match the active parameter count, each user would need LoRA modules of rank 16.

The results are presented in Table 1, where our method outperforms the baseline methods for all *in-distribution* tasks, regardless of matching the full parameter count or the active parameter count. This advantage stems from the fact that both `HetLoRA` and `FlexLoRA` average model parameters across

users without allocating parameters for local adaptations, focusing on building a strong generalist model. In contrast, our approach adaptively integrates both generalist and specialist knowledge, excelling in scenarios where specialized knowledge is crucial.

Table 1: Mean test ppl (std) over users with heterogeneous models, averaged across 3 seeds. Light / dark grey denote in-distribution and out-of-distribution tasks respectively.

| | OURS CoMiGS-1GXS | HetLoRA ACTIVE | FULL | FlexLoRA ACTIVE | FULL |
|---|---|---|---|---|---|
| *GPT2-124M* | | | | | |
| **MULTILINGUAL** | | | | | |
| *(2,2,4,4)* | **46.48 (0.16)** | 57.76 (0.10) | 58.60 (0.20) | 77.71 (0.15) | 77.66 (0.06) |
| *(4,4,2,2)* | **47.24 (0.09)** | 57.76 (0.10) | 59.14 (0.04) | 77.71 (0.15) | 75.64 (0.19) |
| **SLIMPAJAMA** | | | | | |
| *(2,4,4,2)* | **22.10 (0.17)** | 23.33 (0.10) | 23.15 (0.09) | 22.98 (0.10) | 23.03 (0.07) |
| *(4,2,2,4)* | **22.28 (0.09)** | 23.33 (0.10) | 23.17 (0.09) | 22.98 (0.10) | 23.03 (0.08) |
| **AG NEWS** | | | | | |
| *(4,2,2,2)* | 33.66 (0.07) | **31.58 (0.14)** | 31.95 (0.13) | 36.41 (0.18) | 36.62 (0.11) |
| *(2,4,4,4)* | 34.22 (0.09) | **31.58 (0.14)** | 32.52 (0.19) | 36.41 (0.18) | 36.46 (0.04) |
| *Llama3.2-1B* | | | | | |
| **COMMON-CORPUS** | | | | | |
| *(2,4,4,2)* | **18.74 (0.14)** | 21.41 (0.12) | 21.74 (0.09) | 24.63 (0.12) | 25.18 (0.08) |
| *(4,2,2,4)* | **18.68 (0.11)** | 21.41 (0.12) | 21.61 (0.10) | 24.63 (0.12) | 24.74 (0.09) |
| **AG NEWS** | | | | | |
| *(4,2,2,2)* | 16.39 (0.11) | **15.89 (0.05)** | 16.02 (0.05) | 17.33 (0.04) | 17.52 (0.04) |
| *(2,4,4,4)* | 16.44 (0.07) | **15.89 (0.05)** | 16.25 (0.11) | 17.33 (0.04) | 17.70 (0.10) |

Table 2: Mean (std) test perplexity over the users with homogeneous models, averaged across 3 seeds (the lower the better). Light grey denotes in-distribution tasks and dark grey denotes out-of-distrition tasks.

| Base Model | GPT2-124M | | | LLAMA3.2-1B | |
|---|---|---|---|---|---|
| *Dataset* | *Multilingual* | *SlimPajama* | *AG News* | *Com-Corpus* | *AG News* |
| *Pretrained* | 156.12 | 37.19 | 90.65 | 30.40 | 29.37 |
| *Centralized* | 55.41 (0.12) | 19.53 (0.14) | 28.19 (0.52) | 17.97 (0.19) | 16.12 (0.05) |
| *Local* | 54.38 (0.32) | 26.95 (0.14) | 41.46 (0.06) | 20.19 (0.11) | 19.96 (0.01) |
| *FedAvg* | 58.80 (0.34) | 23.27 (0.05) | 31.84 (0.02) | 21.95 (0.11) | 15.86 (0.05) |
| *PCL* | 54.53 (0.19) | 26.99 (0.19) | 32.25 (0.12) | 19.65 (0.03) | 16.84 (0.05) |
| *pFedMoE* | 52.27 (0.17) | 25.40 (0.09) | 38.72 (0.21) | 20.41 (0.05) | 17.84 (0.05) |
| *FDLoRA* | 57.45 (0.81) | 22.71 (0.40) | 33.61 (0.07) | 22.11 (0.05) | 16.64 (0.02) |
| *CoMiGS - 2S* | **46.36 (0.16)** | 22.51 (0.08) | 35.81 (0.13) | 18.46 (0.13) | 18.03 (0.11) |
| *CoMiGS - 2G* | 58.31 (0.17) | **21.36 (0.01)** | **31.18 (0.05)** | 20.18 (0.09) | **15.41 (0.05)** |
| *CoMiGS - 1GIS* | 47.19 (0.10) | 21.79 (0.04) | 33.53 (0.03) | **18.37 (0.03)** | 16.31 (0.05) |

## 3.3 USER-SPECIFIC ANALYSIS

In this section, we investigate how each user can benefit from our CoMiGS-1GXS. In practice, users may not know their local data complexity, leading to a potential mismatch in resource allocation relative to data quantity. To simulate such scenarios, we allocate model capabilities—measured by $n_i$ (the number of LoRA modules per user)—either positively or negatively correlated with their local data size. It is important to note that one generalist is always assigned. Top2 routing is always performed when $n_i \geq 2$. The results are shown in Fig. 5.

**More Specialists Help with Higher Data Quantity.**  High data quantity users (French and Italian) consistently benefit from having more specialists locally, as their test perplexities decrease when the number of specialists increases from 1 to 3 to 7. This suggests that when sufficient local training data is available, adding more specialists leads to improved performance. (Top panel in Fig. 5)

**Generalists Help to Prevent Redundant Specialists from Over-Fitting.**  For users with low data quantities, local model training with just two LoRA modules already results in overfitting (a trend observed in Fig. 7). Our method succeeds to suppress overfitting, even when fine-tuning twice or four times as many expert parameters. We attribute this to the existence of the generalists. (Middle panel in Fig. 5)

**Specialists Can Benefit Generalists.**  What happens if users can only support a maximum of one expert? In our setup, such users must rely on the generalist expert when participating in collaboration. Interestingly, even when their collaborators are allocated more specialists, low-resourced users with only one generalist still benefit from the refined role diversification between generalists and specialists. As a result, the generalists become more powerful. (Bottom panel in Fig. 5)

## 4 CONCLUSIONS

We propose a novel framework for on-device personalized collaborative fine-tuning of LLMs, grounded in an innovative bi-level formulation of the Mixture-of-Experts learning objective. Our fine-grained integration of generalist and specialist expert knowledge achieves superior performance in balancing personalization and collaboration within Federated LLMs.

Furthermore, our framework is the first to address both model and data heterogeneity in collaborative LLM training. It further decouples local data quantity from resource availability, allowing high-resourced users to leverage larger datasets for improved performance while remaining resilient against overfitting in low-data scenarios. CoMiGS is both theoretically sound and resource-efficient for practical deployment.

## IMPACT STATEMENT

We offer a collaboration framework for edge devices, aiming to enable smaller devices to leverage large language models (LLMs) despite limited resources and data availability. Our approach enhances fairness and mitigates privacy concerns by ensuring data remains on end devices. The privacy aspects can further be enhanced by differential private aggregation of generalist weights, which we do not pursue here.

The robustness towards attackers is beyond the scope of our work. Our collaboration framework has no guarantee of resilience towards adversarial attackers through the aggregation of the generalist weights, which could potentially lead to misuse by certain parties. Further research is required on top of our framework to guarantee its safe deployment.

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

# A    OUR ALGORITHM

The pseudo codes of our proposed CoMiGS method are presented in Alg. 1. While the scheme requires a server, it can alternatively be implemented in a serverless all2all fashion, which requires $N$ times more communication overhead and we do not further pursue this here.

---

**Algorithm 1** Pseudo code of our proposed algorithm

---

**Input:** Expert parameters $\{\boldsymbol{\theta}_{i,0}^G, \boldsymbol{\theta}_{i,0}^S\}$, routing parameters $\{\boldsymbol{\phi}_{i,0}\}$. Local training data and validation data $\{\boldsymbol{X}_i^{\text{train}}, \boldsymbol{X}_i^{\text{valid}}\}$, $i \in \{1, 2, .., N\}$. Communication round $T$ and routing update period $\tau$. Load balancing weight $\lambda$.

**for** $t = 1, ..., T$ **do**

    Server aggregates generalist parameters: $\boldsymbol{\theta}_{t-1}^G = \frac{1}{N} \sum_i \boldsymbol{\theta}_{i,t-1}^G$

    **for** $i \in [0, N)$ **do**

        Users download aggregated generalist weights and

        prepare model parameters for training $\{\boldsymbol{\theta}_{t-1}^G, \boldsymbol{\theta}_{i,t-1}^S, \boldsymbol{\phi}_{i,t-1}\}$

        Do gradient steps on $(\boldsymbol{\theta}_{t-1}^G, \boldsymbol{\theta}_{i,t-1}^S)$ towards minimizing (3) and get $(\boldsymbol{\theta}_{i,t}^G, \boldsymbol{\theta}_{i,t}^S)$

$$
\min_{\boldsymbol{\theta}_i^G, \boldsymbol{\theta}_i^S} \mathcal{L}(f(\boldsymbol{X}_i^{\text{train}}; \boldsymbol{\theta}_i^G, \boldsymbol{\theta}_i^S, \boldsymbol{\phi}_{i,t-1}), \boldsymbol{X}_i^{\text{train}}) +
$$
$$
\lambda \cdot \mathcal{L}_i^{\text{LB}}(\boldsymbol{X}_i^{\text{train}}; \boldsymbol{\theta}_i^G, \boldsymbol{\theta}_i^S, \boldsymbol{\phi}_{i,t-1}) \tag{3}
$$

        **if** $t\%\tau = 0$ **then**

            Do gradient steps on $\boldsymbol{\phi}_{i,t-1}$ towards minimizing (4) and get $\boldsymbol{\phi}_{i,t}$

$$
\min_{\boldsymbol{\phi}_i} \mathcal{L}(f(\boldsymbol{X}_i^{\text{valid}}; \boldsymbol{\theta}_{i,t}^G, \boldsymbol{\theta}_{i,t}^S, \boldsymbol{\phi}_i), \boldsymbol{X}_i^{\text{valid}}) +
$$
$$
\lambda \cdot \mathcal{L}_i^{\text{LB}}(\boldsymbol{X}_i^{\text{valid}}; \boldsymbol{\theta}_{i,t}^G, \boldsymbol{\theta}_{i,t}^S, \boldsymbol{\phi}_i) \tag{4}
$$

        **end if**

    **end for**

    Each device $i \in \{1, 2, .., N\}$ sends generalist weights $\boldsymbol{\theta}_{i,t}^G$ to the server

**end for**

**Return:** Expert parameters $\{\boldsymbol{\theta}_{i,T}^G, \boldsymbol{\theta}_{i,T}^S\}$ and routing parameters $\{\boldsymbol{\phi}_{i,T}\}$

---

# B    EXPERIMENTAL DETAILS

## B.1    SETUP

### B.1.1    DATASETS

We selected the following datasets to demonstrate the efficacy of our proposed algorithm: 1) *Multilingual Wikipedia*: Wikipedia articles in four languages: German, French and Italian from Wikimedia-Foundation, and Dutch from Guo et al. (2020); 2) *SlimPajama*: We pick the following four categories – StackExchange, Github Codes, ArXiv, Book from Soboleva et al. (2023); 3) *AG News*: News from categories of World, Sports, Business, and Sci/Tech (Zhang et al., 2016). 4) *Common Corpus* (pleias, 2024): specifically the following three categories – YouTube-Commons, Public Domain Books, and EU Tenders collections, and the Harvard US Patent dataset from Suzgun et al. (2022).

The number of tokens for our experiments within each user is shown in Table 3.

Given the extensive pre-training of Llama 3.2 models on over 15 trillion tokens from public sources, and the multilingual capabilities of Llama 3.2 - 1B, fine-tuning on multilingual Wikipedia or SlimPajama resulted in negligible improvements likely due to significant overlap with the pre-training data corpus. We curated another more difficult fine-tuning dataset – Common Corpus to show case the distinctions of the baseline methods.

Table 3: Number of tokens in each dataset splits

|  |  | User 1 | User 2 | User 3 | User 4 |
|---|---|---|---|---|---|
| **Multilingual** | TRAINING | 557'662 | 407'498 | 556'796 | 451'584 |
|  | VALIDATION | 300'764 | 216'318 | 220'071 | 165'984 |
|  | TEST | 229'720 | 219'741 | 210'570 | 172'547 |
| **SlimPajama** | TRAINING | 1'000'000 | 1'000'000 | 1'000'000 | 1'000'000 |
|  | VALIDATION | 200'000 | 200'000 | 200'000 | 200'000 |
|  | TEST | 200'000 | 200'000 | 200'000 | 200'000 |
| **AG News** | TRAINING | 761'924 | 756'719 | 814'131 | 771'460 |
|  | VALIDATION | 48'809 | 48'730 | 50'398 | 48'249 |
|  | TEST | 48'167 | 47'721 | 48'344 | 49'377 |
| **Common Corpus** | TRAINING | 1'000'000 | 1'000'000 | 1'000'000 | 1'000'000 |
|  | VALIDATION | 200'000 | 200'000 | 200'000 | 200'000 |
|  | TEST | 200'000 | 200'000 | 200'000 | 200'000 |

### B.1.2 EXPERIMENTAL DETAILS

We choose the following base model architectures: GPT2 (124M, English only) and Llama 3.2(1B, Multilingual)[1]. We incorporate LoRA modules into every linear layer, including MLP and Self-Attention Layers, following the recommendations of Fomenko et al. (2024). A routing mechanism is exclusively implemented atop MLP layers. The number of LoRA experts in MLP blocks depends on the local resource abundance.

For training, we followed Kalajdzievski (2023). We choose $\gamma$ to be a rank-stabilized value, a technique which helps stabilize gradient norms. $\alpha$ and the rank $r$ are hyper-parameters to choose from. The LoRA modules function as follows:

$$W = W^0 + \gamma \cdot AB, \qquad \gamma = \frac{\alpha}{\sqrt{r}} \tag{5}$$

All our experiments except the centralized ones were conducted on a single A100-SXM4-40GB GPU. The centralized learning baseline experiments were conducted on a single A100-SXM4-80GB GPU, as a batch size of 64*4 requires a larger storage capacity.

We use a constant learning rate of $2 \times 10^{-3}$ for updating routing parameters and a $2 \times 10^{-3}$ learning rate with a one-cycle cosine schedule for expert parameters during fine-tuning. The LoRA rank $r$ is set to 8 unless otherwise specified, with LoRA alpha $\alpha$ set to 16, following the common practice of setting alpha to twice the rank (Raschka, 2023). A load balancing weight $0.01$ is always applied.

**GPT2 Experiments.** For AG News and Multilingual Wikipedia data splits, we conduct 20 communication rounds. For SlimPajama data splits, due to greater category diversity, we conduct 50 communication rounds. Between each pair of communication rounds, there are 10 local iterations. In each iteration, a batch size of 64 is processed with a context length of 128. We set the routing update period to 30 iterations, and every time we update routing parameters, we do 10 gradient steps on the validation loss. The choice of the hyperparamters is from a sweep run and we provide the evidence in Fig. 6.

**Llama3.2 Experiments.** For AG News data splits, we conduct 10 communication rounds. For Common-corpus data splits, due to greater category diversity, we conduct 20 communication rounds. Between each pair of communication rounds, there are 10 local iterations. In each iteration, a batch size of 64 is processed with a context length of 128. We set the routing update period to 30 iterations, and every time we update routing parameters, we do 10 gradient steps on the validation loss.

---

[1]We adopt the codes from `https://github.com/karpathy/nanoGPT` and `https://github.com/danielgrittner/nanoGPT-LoRA`, `https://github.com/pjlab-sys4nlp/llama-moe`

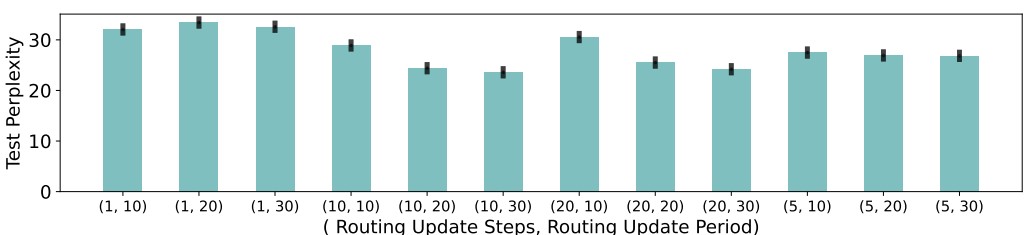

Figure 6: Sweep results on SlimPajama data splits using GPT2-124M base model. We ablate the impact of the update period ($\tau$) and the number of update steps ($s$) on model performance.

## C  COMPUTATIONAL AND COMMUNICATION OVERHEAD

Our approach offers a significant advantage for on-device deployment due to its minimal computational and communication overhead. We compare the resource consumption of our `CoMiGS-1G1S` to `FedAvg` in Table 4, matching the parameter count for LoRA modules.

The communication costs are halved compared to standard `FedAvg`, as only the weights of generalist experts are exchanged. Our framework employs a first-order algorithm, ensuring that computation and memory requirements remain on par with those of standard `FedAvg` algorithms. The additional memory and computational overhead primarily stem from the inclusion of the router, which is minimal (1.25% increase) since the router consists of a single-layer MLP.

Table 4: Extra resource consumption (per device) `CoMiGS-1G1S` compared to standard `FedAvg`, assuming base model is GPT-124M with bfloat16 training.

| COMP. OVERHEAD / FORWARD PASS | MEMORY | COMM. COSTS / ROUND |
|---|---|---|
| + 5 MFLOPS (+1.25%) | + 0.035 MB (+1.25%) | -1.41 MB (-50%) |

## D  MORE TABLES AND FIGURES

### D.1  LEARNING CURVES OF DIFFERENT METHODS

See Fig. 7.

### D.2  EXTENDED BASELINE COMPARISON

An extended version of Table 2 is presented in Table 5. In this extension, we incorporate two additional ablations: 1) Integration of a routing mechanism, updated simultaneously with the expert networks; 2) Iterative updates alternating between routing and expert parameters, with the routing parameters updated using newly-sampled training batches instead of a dedicated validation set. 2) is to address the scenario where a validation set is not available.

Moreover, we include two other baseline methods – `FFA-LoRA` from Sun et al. (2024) and `FedSA` from Guo et al. (2024). `FFA-LoRA` keeps the LoRA A matrices fixed at initialization, while `FedSA` always aggregates LoRA A matrices but leave LoRA B matrices localized.

Notably, the comparison between scenarios ii) and iii) reveals minimal disparity, underscoring the significance of having an independent validation set exclusively for routing parameter updates.

### D.3  HETLORA

Analogously to the baseline experiment comparison in FlexLoRA (Bai et al., 2024), we use $\gamma = 0.99$ as pruning strength and sweep the regularization parameter in $\{5 \times 10^{-2}, 5 \times 10^{-3}, 5 \times 10^{-4}\}$.

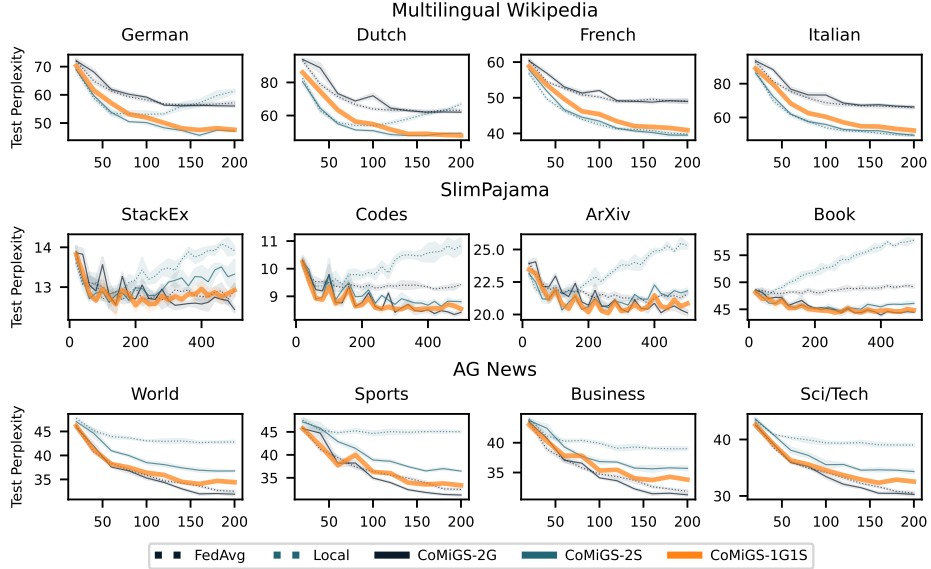

Figure 7: Test Perplexity during training (base model: GPT2-124M): our method closely follows the best performing method

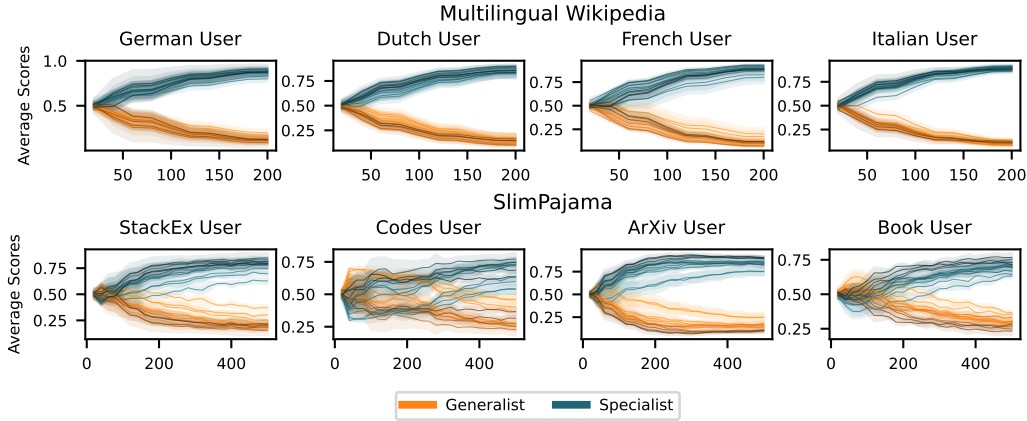

Figure 8: Expert Scores for the *generalist* expert and the *specialist* expert from our `CoMiGS-1G1S` method, averaged across all tokens and multiple batches for the in-distribution task, with x-axis being the number of iterations. Darker colors represent deeper layers.

## D.4 IS THE STANDARD LOAD BALANCING LOSS SUFFICIENT?

The standard load balancing loss encourages equal assignment of tokens to each expert. When the number of experts gets larger, there might not be enough tokens routed to the generalists, which might lead to a under-developed general knowledge. We will verify if this is indeed true.

To encourage enough tokens to be routed to the generalist expert such that more general knowledge can be developed, we modify our load-balancing loss by introducing importance weighting. As we separate the 0-th expert to be the generalist expert and conduct Top-2 routing, the modified load balancing loss is as follows:

$$\mathcal{L}_i^{\text{LB}} = \frac{1}{(n_i-1)^2+1} \cdot f_0 \cdot P_0 + \sum_{j=1}^{n_i-1} \frac{n_i-1}{(n_i-1)^2+1} \cdot f_j \cdot P_j \tag{6}$$

Table 5: Mean test perplexity over users with homogenous models, averaged across 3 seeds. Mean (std) with a rank locator for the mean (the lower the better). Green denotes the best performing methods and red denotes our method.

| | IN DISTRIBUTION | | OUT OF DISTRIBUTION |
| | *Multilingual* | *SlimPajama* | *AG News* |
| --- | --- | --- | --- |
| I) WITHOUT ROUTING | | | |
| *Pretrained* | 156.12 | 37.19 | 90.65 |
| *Centralized* | 55.41 (0.12) | 19.53 (0.14) | 28.19 (0.52) |
| *Local* | 54.38 (0.32) | 26.95 (0.14) | 41.46 (0.06) |
| *FedAvg* | 58.80 (0.34) | 23.27 (0.05) | 31.84 (0.02) |
| *FFA-LoRA* | 66.80 (0.20) | 22.85 (0.12) | 33.13 (0.09) |
| *FedSa-LoRA* | 57.60 (0.14) | 23.40 (0.13) | 31.57 (0.10) |
| *PCL* | 54.53 (0.19) | 26.99 (0.19) | 32.25 (0.12) |
| II) UPDATE ROUTING AND EXPERT PARAMS SIMULTANEOUSLY ON TRAINING LOSS | | | |
| *Local-MoE* | 55.27 (0.40) | 27.16 (0.16) | 41.49 (0.01) |
| *FedAvg-MoE* | 56.77 (0.37) | 23.32 (0.07) | 32.24 (0.08) |
| *pFedMoE* | 52.27 (0.17) | 22.91 (0.18) | 38.72 (0.21) |
| III) ALTERNATING UPDATE ROUTING PARAMS ON NEWLY SAMPLED BATCHES FROM TRAINING SET | | | |
| *Local-MoE - tr* | 53.78 (0.33) | 27.78 (0.06) | 41.46 (0.03) |
| *FedAvg-MoE - tr* | 59.39 (0.13) | 23.00 (0.01) | 31.70 (0.16) |
| *CoMiGS - tr* | 50.86 (0.14) | 25.45 (0.01) | 38.93 (0.08) |
| IV) ALTERNATING UPDATE ROUTING PARAMS ON A VALIDATION SET | | | |
| *CoMiGS - 2S* | 46.36 (0.16) | 22.51 (0.08) | 35.81 (0.13) |
| *CoMiGS - 2G* | 58.31 (0.17) | 21.36 (0.01) | 31.18 (0.05) |
| *CoMiGS - 1G1S* | 47.19 (0.10) | 21.79 (0.04) | 33.53 (0.03) |

Table 6: Test perplexity with different load balancing terms with (hetero) or without (homo) resource heterogeneity.

| | No LB | LB (uniform) | LB (generalist-favored) |
| --- | --- | --- | --- |
| AG News (homo) | 33.69 (0.21) | 33.53 (0.03) | 33.53 (0.03) |
| AG News (hetero) | 34.31 (0.05) | 34.28 (0.11) | 34.22 (0.09) |
| Multi-Wiki (homo) | 47.31 (0.15) | 47.19 (0.10) | 47.19 (0.10) |
| Multi-Wiki (hetero) | 46.36 (0.16) | 46.15 (0.04) | 46.48 (0.16) |
| SlimPajama (homo) | 21.77 (0.02) | 21.79 (0.04) | 21.79 (0.04) |
| SlimPajama (hetero) | 22.15 (0.07) | 22.10 (0.11) | 22.10 (0.17) |

where

$$f_j = \frac{1}{T} \sum_{x \in \mathcal{B}} \mathbb{1}\{j \in \text{Top2 indices of } p(x)\} \qquad P_j = \frac{1}{T} \sum_{x \in \mathcal{B}} p_j(x) \qquad (7)$$

$j$ is the expert index and $p(x) = [p_j(x)]_{j=1}^{n_i}$ is the logit output from the routing network for a specific token $x$. The idea is that one of the top 2 tokens should always be routed to the generalist expert, i.e. the 0-th expert. Thus, $\frac{p_0}{1/2}$ should be equal to $\frac{p_i}{1/2(n_i-1)}$ for $i \neq 0$. As the original load balancing loss encourages uniform distribution, this modification encourages the generalist expert to have a routing probability of 0.5 on expectation. Note that when $n_i = 2$, this $\mathcal{L}_i^{\text{LB}}$ is the same as the original load balancing loss as proposed in Fedus et al. (2022).

We present the results in Table 6: in both scenarios, whether users have the same or different numbers of experts, including a load-balancing term leads to a slight improvement compared to omitting it. However, encouraging more tokens to be routed to the generalists does not make a significant difference.

# E    ADDITIONAL EXPERIMENTS

We replicate the experiments in Section 3.2 with the SlimPajama dataset, where we assign four times as many tokens to ArXiv User and Book User as to Stack Exchange User and Codes User.

**More Specialists Help with Higher Data Quantity.** From Fig. 9, it is evident that ArXiv User and Book User, with abundant local data, benefit from having more local experts.

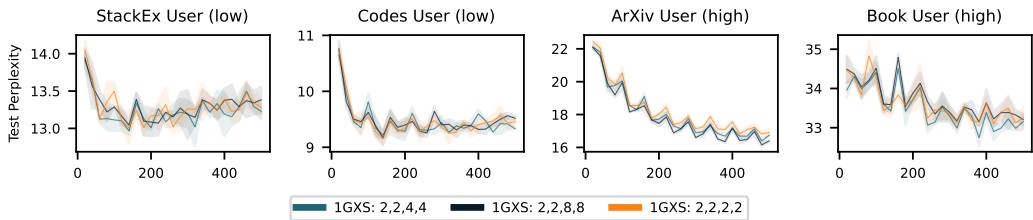

Figure 9: Test Perplexity during training for the SlimPajama setup. ArXiv User and Book User have more local data and thus benefit from having more experts. The numbers in the legend indicate the number of experts $n_i$ within each user. Top-2 routing is performed.

**Generalists Help to Prevent Redundant Specialists from Over-Fitting?** From Fig. 10, we observe more prominent overfitting than in Fig. 5, likely because the tasks are objectively easier, as indicated by lower test perplexity from the beginning of fine-tuning. Generalists have limited power to prevent overfitting with easy tasks.

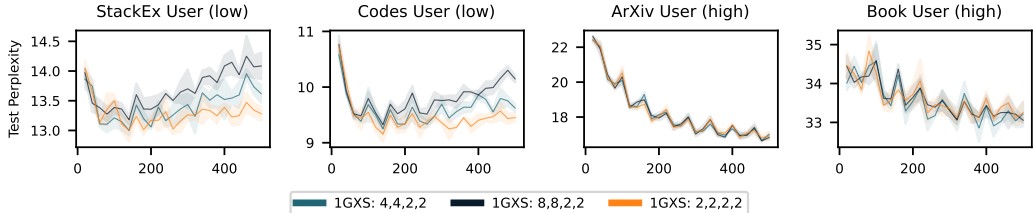

Figure 10: In this SlimPajama setup, Stack Ex User and Codes User despite having low resources locally, overfit slightly on their small-sized local data. Numbers in the legend denote the number of experts $n_i$ within each user. Top2 routing is performed.

**Specialists Can Benefit Generalists.** Low-resourced users that can only support a single expert setup still benefit from collaboration, as the generalist knowledge is refined through a more detailed distinction between specialist and generalist roles via other high-resourced users. This is indicated by the enhanced performances for Stack Exchange and Codes Users.

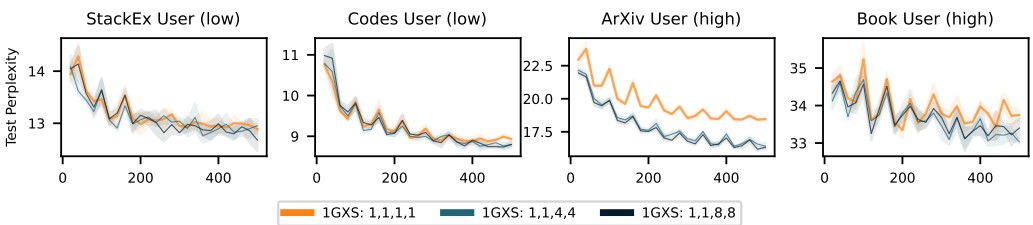

Figure 11: In this SlimPajama setup, Stack Ex User and Codes User, despite having only one expert locally, still benefit from other users having more experts, thereby enhancing the generalist's performance. The numbers in the legend indicate the number of experts, $n_i$, within each user. Top-2 routing is applied when $n_i \geq 2$

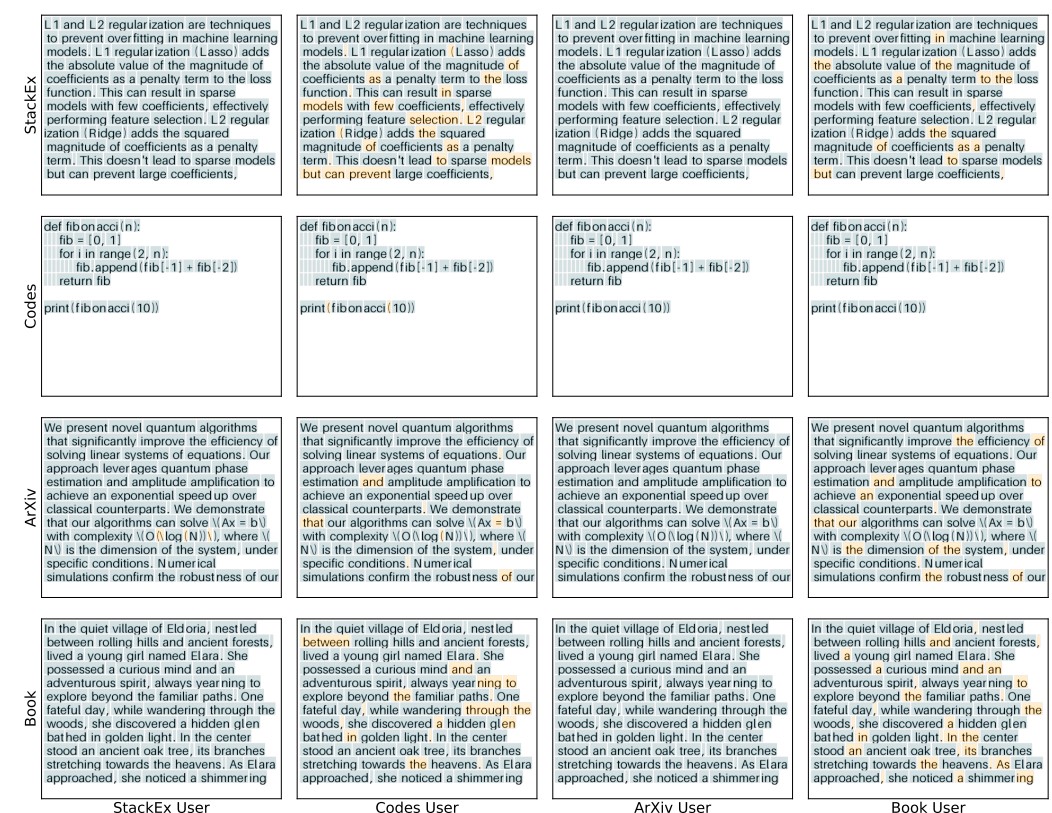

Figure 12: Visualization of token-level routing results for `CoMiGS-1G1S` trained on SlimPajama. Tokens are colored with the first expert choice at the 0th (first) layer. Orange denotes the generalist and blue denotes the specialist. Diagonal entries are in-distribution texts and off-diagonal entries are out-of-distribution texts. Texts are generated by ChatGPT.

## F    VISUALIZATION OF EXPERT SPECIALIZATION

To visualize which tokens are routed to the generalist and specialist experts for our `CoMiGS-1G1S` model trained on SlimPajama, we ask ChatGPT to generate texts in the style of StackExchange, Python Codes, ArXiv Paper and Books. We then feed those texts to the user-specific models and color the token with the Top1 routed index. The routing results after the very first layer (0th), a middle layer (5th), and the very last layer (11th) are presented in Fig. 12, 13 and 14.

We perform the same experiments on AG News, asking ChatGPT to generate News text on the topics World, Sports, Business, and Sci/Tech. The routing results after the very first layer (0th), a middle layer (5th), and the very last layer (11th) are presented in Fig. 15, 16 and 17.

For all the plots, diagonal entries are *in-distribution* texts and off-diagonal entries are *out-of-distribution* texts.

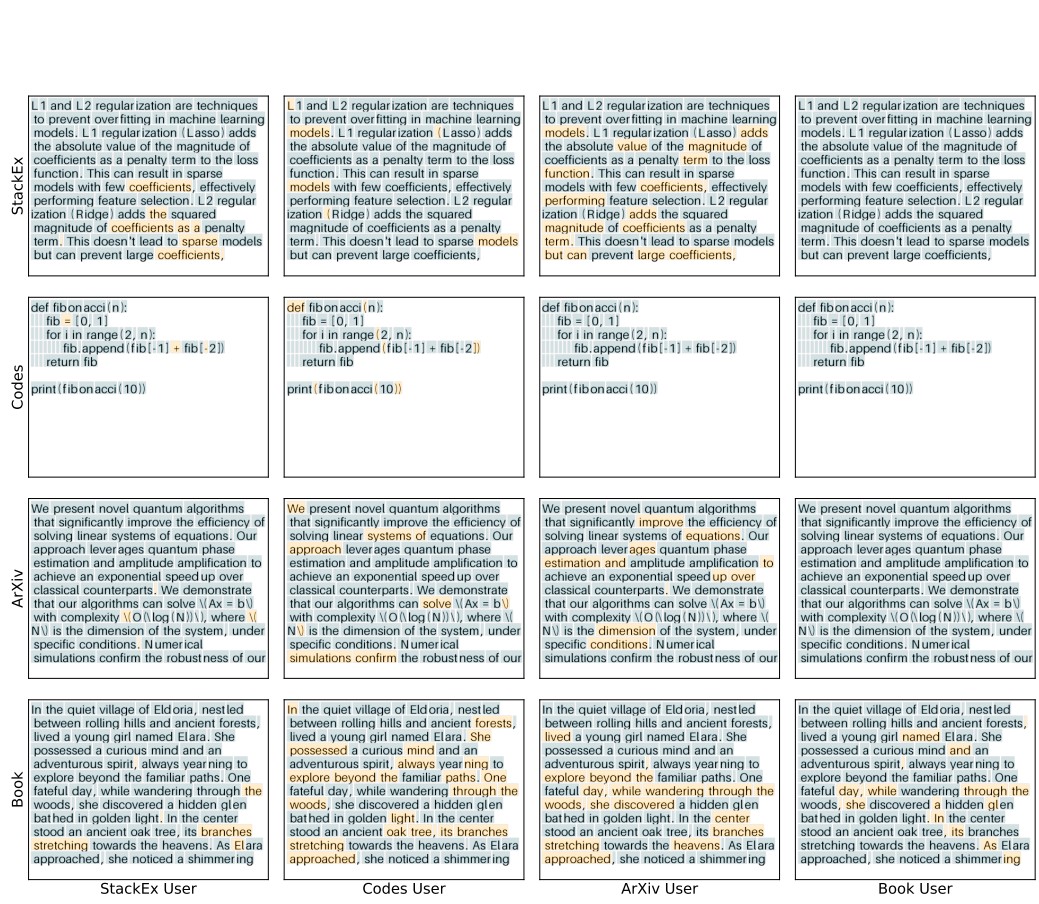

Figure 13: Visualization of token-level routing results for `CoMiGS-1G1S` trained on SlimPajama. Tokens are colored with the first expert choice at the 5th layer. Orange denotes the generalist and blue denotes the specialist. Diagonal entries are in-distribution texts and off-diagonal entries are out-of-distribution texts. Texts are generated by ChatGPT.

918
919
920
921
922
923
924
925
926
927
928
929
930
931
932
933
934
935
936
937
938
939
940
941
942
943
944
945
946
947
948
949
950
951
952
953
954
955
956

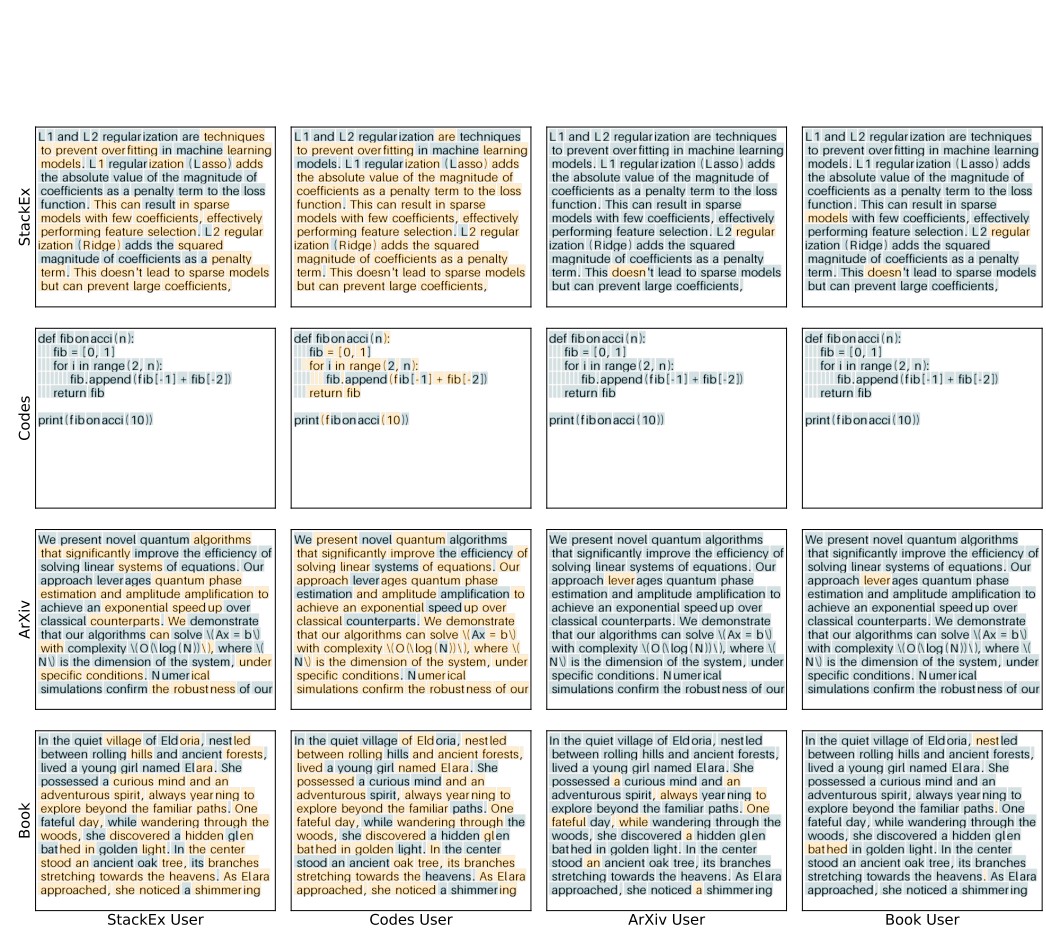

Figure 14: Visualization of token-level routing results for `CoMiGS-1G1S` trained on SlimPajama. Tokens are colored with the first expert choice at the 11th (last) layer. Orange denotes the generalist and blue denotes the specialist. Diagonal entries are in-distribution texts and off-diagonal entries are out-of-distribution texts. Texts are generated by ChatGPT.

957
958
959
960
961
962
963
964
965
966
967
968
969
970
971

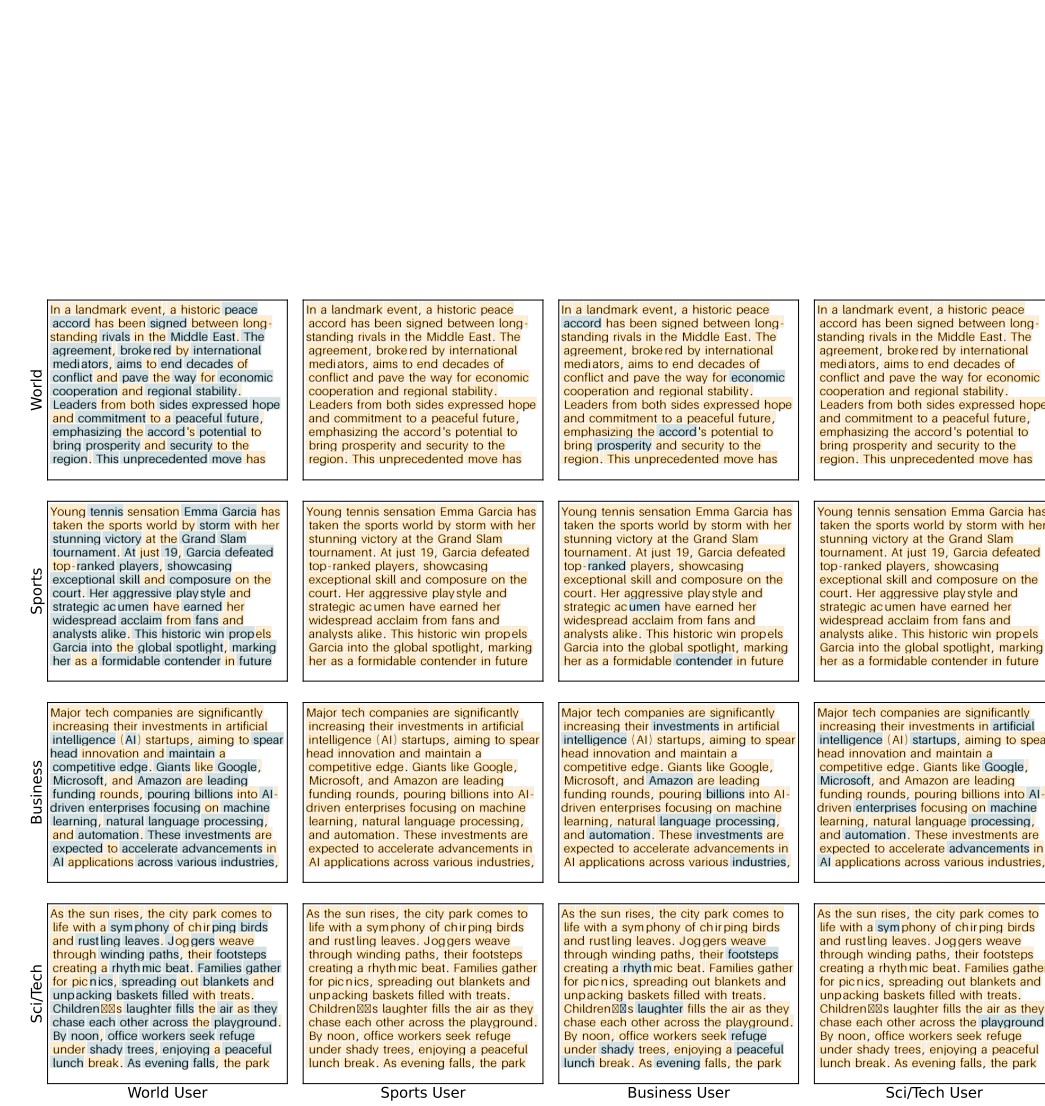

Figure 15: Visualization of token-level routing results for `CoMiGS-1G1S` trained on AG News. Tokens are colored with the first expert choice at the 0th (first) layer. Orange denotes the generalist and blue denotes the specialist. Diagonal entries are in-distribution texts and off-diagonal entries are out-of-distribution texts. Texts are generated by ChatGPT.

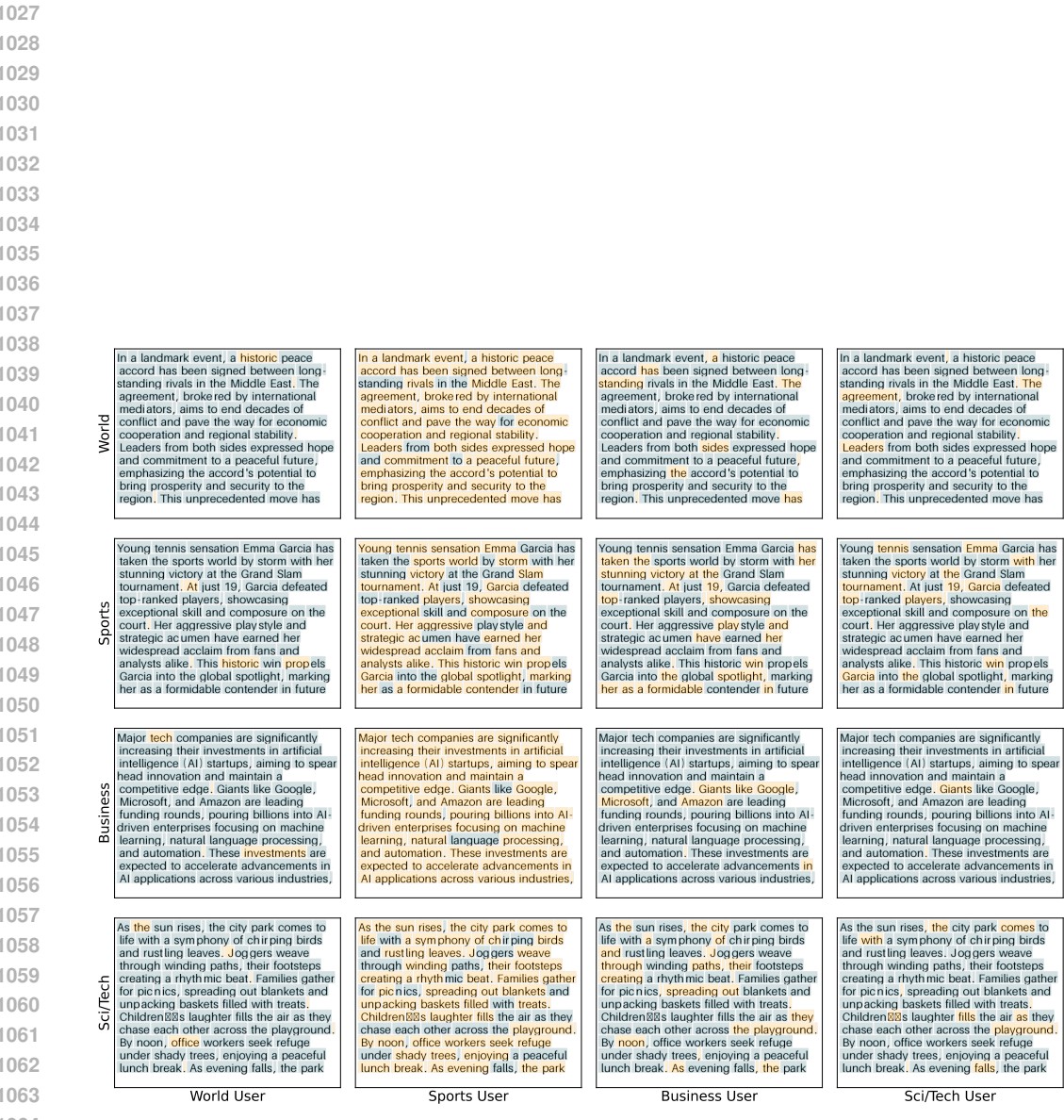

Figure 16: Visualization of token-level routing results for `CoMiGS-1G1S` trained on AG News. Tokens are colored with the first expert choice at the 5th (middle) layer. Orange denotes the generalist and blue denotes the specialist. Diagonal entries are in-distribution texts and off-diagonal entries are out-of-distribution texts. Texts are generated by ChatGPT.

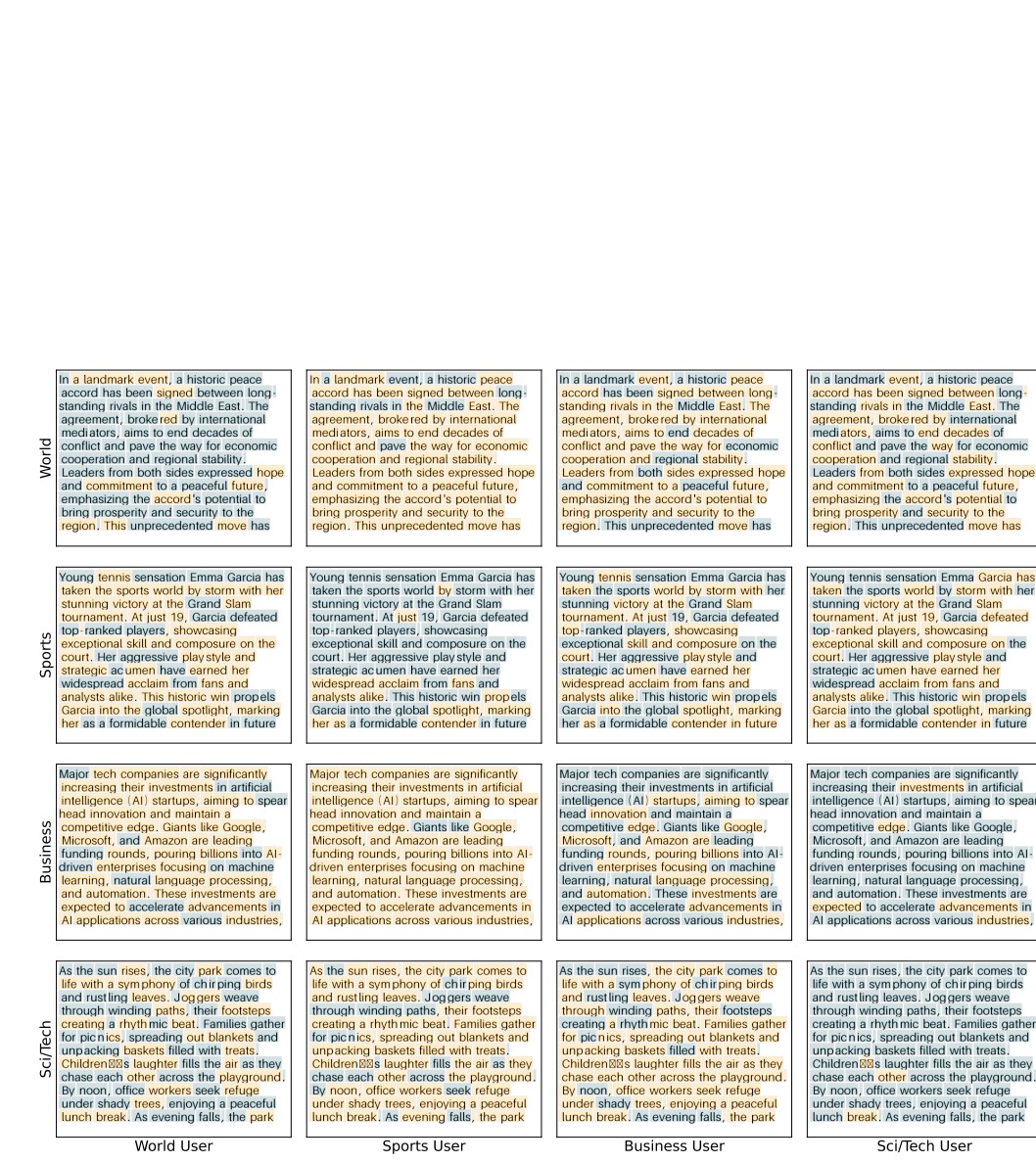

Figure 17: Visualization of token-level routing results for `CoMiGS-1G1S` trained on AG News. Tokens are colored with the first expert choice at the 11th (last) layer. Orange denotes the generalist and blue denotes the specialist. Diagonal entries are in-distribution texts and off-diagonal entries are out-of-distribution texts. Texts are generated by ChatGPT.

# G  ALTERNATING MINIMIZATION CONVERGENCE

## G.1  NOTATION

Let $f_1(\boldsymbol{\Theta}, \boldsymbol{\Phi}) \equiv f_{\text{valid}}(\boldsymbol{\Theta}, \boldsymbol{\Phi})$ and $f_2(\boldsymbol{\Theta}, \boldsymbol{\Phi}) \equiv f_{\text{train}}(\boldsymbol{\Theta}, \boldsymbol{\Phi})$. We denote partial minimization operators from (1) by

$$
\begin{aligned}
\boldsymbol{\Phi}_{k+1} &= \underset{\boldsymbol{\Phi} \in \Omega}{\arg\min}\, f_1(\boldsymbol{\Theta}_k, \boldsymbol{\Phi}), \\
\boldsymbol{\Theta}_{k+1} &= \underset{\boldsymbol{\Theta} \in Q}{\arg\min}\, f_2(\boldsymbol{\Theta}, \boldsymbol{\Phi}_{k+1}).
\end{aligned}
\tag{8}
$$

and their compositions by $T := u_2 \circ u_1$ and $P := u_1 \circ u_2$. Note that both $T$ and $P$ act on the corresponding spaces of $\boldsymbol{\Theta}$ and $\boldsymbol{\Phi}$: $T : \mathbb{R}^{|\boldsymbol{\Theta}|} \to \mathbb{R}^{|\boldsymbol{\Theta}|}$ and $P : \mathbb{R}^{|\boldsymbol{\Phi}|} \to \mathbb{R}^{|\boldsymbol{\Phi}|}$.

**Assumption 1** (Shared Optima). *There exist $\boldsymbol{\Theta}^\star$ and $\boldsymbol{\Phi}^\star$ such that*

$$
\boldsymbol{\Theta}^\star = T(\boldsymbol{\Theta}^\star) \qquad and \qquad \boldsymbol{\Phi}^\star = P(\boldsymbol{\Phi}^\star).
\tag{9}
$$

**Remark 1.** *Eq. (9) means that $f_{valid}$ and $f_{train}$ share the same global optima, which is reasonable when the train and validation data are similar, $\boldsymbol{X}_i^{train} \sim \boldsymbol{X}_i^{valid}$, and, hence, $f_{valid} \approx f_{train}$. It also holds for overparametrized models, such as LLMs.*

## G.2  CONTRACTION AND CONVERGENCE

As we will see, it is natural to assume that operators $u_1$ and $u_2$ are *contractions*. We will provide a working example of our setting in the next section, where this condition will hold. We assume to have some norms fixed on $Q$ and $\Omega$, that are not necessarily Euclidean. For simplicity, and when it is clear from the context, we will use the same symbol $\|\cdot\|$ for both norms, even though they can be different for spaces of $\boldsymbol{\Theta}$ and $\boldsymbol{\Phi}$.

**Assumption 2** (Contraction Property). *Let $u_1$ and $u_2$ be Lipschitz with some constants $\lambda_1, \lambda_2 > 0$, for any $\boldsymbol{\Theta}, \bar{\boldsymbol{\Theta}}$ and $\boldsymbol{\Phi}, \bar{\boldsymbol{\Phi}}$:*

$$
\begin{aligned}
\|u_1(\boldsymbol{\Theta}) - u_1(\bar{\boldsymbol{\Theta}})\| &\leq \lambda_1 \|\boldsymbol{\Theta} - \bar{\boldsymbol{\Theta}}\|, \\
\|u_2(\boldsymbol{\Phi}) - u_2(\bar{\boldsymbol{\Phi}})\| &\leq \lambda_2 \|\boldsymbol{\Phi} - \bar{\boldsymbol{\Phi}}\|.
\end{aligned}
\tag{10}
$$

Under these assumptions we can show the convergence of the sequence $\{\boldsymbol{\Theta}_k\}_{k \geq 0}$. Indeed, for every $k \geq 0$, we have

$$
\|\boldsymbol{\Theta}_{k+1} - \boldsymbol{\Theta}^\star\| = \|T(\boldsymbol{\Theta}_k) - \boldsymbol{\Theta}^\star\| \overset{\text{Assump.1}}{=} \|T(\boldsymbol{\Theta}_k) - T(\boldsymbol{\Theta}^\star)\|
$$

$$
= \|u_2(u_1(\boldsymbol{\Theta}_k)) - u_2(u_1(\boldsymbol{\Theta}^\star))\| \overset{(9)}{\leq} \lambda_2 \|u_1(\boldsymbol{\Theta}_k) - u_1(\boldsymbol{\Theta}^\star)\| \overset{(10)}{\leq} \lambda_1 \lambda_2 \|\boldsymbol{\Theta}_k - \boldsymbol{\Theta}^\star\|,
$$

and we see that $\boldsymbol{\Theta}_k \to \boldsymbol{\Theta}^\star$ with the linear rate. The same reasoning can be applied to the sequence $\{\boldsymbol{\Phi}_k\}_{k \geq 1}$. Thus, we have established the following general convergence result.

**Theorem G.1** (Theorem 2.1). *Let Assumptions 1, 2 hold and $\lambda_1 \cdot \lambda_2 < 1$. Then, the sequence $(\boldsymbol{\Theta}_k, \boldsymbol{\Phi}_k)_{k \geq 0}$ generated by alternating process (8) converges to $(\boldsymbol{\Theta}^\star, \boldsymbol{\Phi}^\star)$ linearly, for every $k \geq 0$:*

$$
\begin{aligned}
\|\boldsymbol{\Theta}_k - \boldsymbol{\Theta}^\star\| &\leq (\lambda_1 \lambda_2)^k \|\boldsymbol{\Theta}_0 - \boldsymbol{\Theta}^\star\|, \\
\|\boldsymbol{\Phi}_k - \boldsymbol{\Phi}^\star\| &\leq (\lambda_1 \lambda_2)^k \|\boldsymbol{\Phi}_0 - \boldsymbol{\Phi}^\star\|.
\end{aligned}
\tag{11}
$$

**Example 1.** *Consider the following quadratic objective*

$$
f(\boldsymbol{\Theta}, \boldsymbol{\Phi}) = \tfrac{1}{2}\langle \boldsymbol{A}\boldsymbol{\Theta}, \boldsymbol{\Theta}\rangle + \tfrac{1}{2}\langle \boldsymbol{B}\boldsymbol{\Phi}, \boldsymbol{\Phi}\rangle + \langle \boldsymbol{C}\boldsymbol{\Theta}, \boldsymbol{\Phi}\rangle,
$$

*where $\boldsymbol{A} = \boldsymbol{A}^\top \in \mathbb{R}^{|\boldsymbol{\Theta}| \times |\boldsymbol{\Theta}|}$ and $\boldsymbol{B} = \boldsymbol{B}^\top \in \mathbb{R}^{|\boldsymbol{\Phi}| \times |\boldsymbol{\Phi}|}$ are symmetric matrices, and $\boldsymbol{C} \in \mathbb{R}^{|\boldsymbol{\Phi}| \times |\boldsymbol{\Theta}|}$. We assume that $f$ is strictly convex, which means*

$$
\boldsymbol{H} = \begin{bmatrix} \boldsymbol{A} & \boldsymbol{C}^\top \\ \boldsymbol{C} & \boldsymbol{B} \end{bmatrix} \succ \boldsymbol{0}.
$$

*Clearly, for this objective, we have $\boldsymbol{\Theta}^\star = \boldsymbol{0}$ and $\boldsymbol{\Phi}^\star = \boldsymbol{0}$. Then*

$$u_1(\boldsymbol{\Theta}) \;\; := \;\; \arg\min_{\boldsymbol{\Phi}} f(\boldsymbol{\Theta}, \boldsymbol{\Phi}) \;\; = \;\; -\boldsymbol{B}^{-1}\boldsymbol{C}\boldsymbol{\Theta} \qquad \textit{and}$$

$$u_2(\boldsymbol{\Phi}) \;\; := \;\; \arg\min_{\boldsymbol{\Theta}} f(\boldsymbol{\Theta}, \boldsymbol{\Phi}) \;\; = \;\; -\boldsymbol{A}^{-1}\boldsymbol{C}^\top\boldsymbol{\Phi}.$$

*Hence, the composition operator $T := u_2 \circ u_1$ is linear:*

$$T(\boldsymbol{\Theta}) \;\; = \;\; \boldsymbol{A}^{-1}\boldsymbol{C}^\top\boldsymbol{B}^{-1}\boldsymbol{C}\boldsymbol{\Theta}, \tag{12}$$

*and it holds*

$$\|T(\boldsymbol{\Theta}) - \boldsymbol{\Theta}^\star\| \;\; \leq \;\; \|\boldsymbol{A}^{-1}\boldsymbol{C}^\top\boldsymbol{B}^{-1}\boldsymbol{C}\| \cdot \|\boldsymbol{\Theta} - \boldsymbol{\Theta}^\star\|.$$

*Now, denoting by $\mu > 0$ and $L \geq \mu$ the smallest and the largest eigenvalues of matrix $\boldsymbol{H}$ correspondingly, and using the Schur complement, we conclude that*

$$\mu\boldsymbol{I} \;\; \preceq \;\; \boldsymbol{A} \;\; \preceq \;\; L\boldsymbol{I}, \qquad \textit{and} \qquad \mu\boldsymbol{I} \;\; \preceq \;\; \boldsymbol{A} - \boldsymbol{C}^\top\boldsymbol{B}^{-1}\boldsymbol{C} \;\; \preceq \;\; L\boldsymbol{I}, \tag{13}$$

*from which we are able to bound the norm of our matrix as follows:*

$$\|\boldsymbol{A}^{-1}\boldsymbol{C}^\top\boldsymbol{B}^{-1}\boldsymbol{C}\| \;\; = \;\; \|\boldsymbol{A}^{-1/2}(\boldsymbol{C}^\top\boldsymbol{B}^{-1}\boldsymbol{C})\boldsymbol{A}^{-1/2}\| \;\; \overset{(13)}{\leq} \;\; \tfrac{L-\mu}{L} \;\; < \;\; 1,$$

*which proves the contraction property.*

**Example 2.** *Note that for a general differentiable function $f$, using the Taylor expansion, the operator $T = u_2 \circ u_1$, where $u_1(\boldsymbol{\Theta}) := \arg\min_{\boldsymbol{\Phi}} f(\boldsymbol{\Theta}, \boldsymbol{\Phi})$ and $u_2(\boldsymbol{\Phi}) := \arg\min_{\boldsymbol{\Theta}} f(\boldsymbol{\Theta}, \boldsymbol{\Phi})$, can be expressed as follows (compare with (12)):*

$$T(\boldsymbol{\Theta}) - \boldsymbol{\Theta}^\star \;\; = \;\; \boldsymbol{H}_{11}^{-1}\boldsymbol{H}_{12}\boldsymbol{H}_{22}^{-1}\boldsymbol{H}_{21}(\boldsymbol{\Theta} - \boldsymbol{\Theta}^\star),$$

*where*

$$\boldsymbol{H}_{11} \;\; = \;\; \int_0^1 \tfrac{\partial^2 f}{\partial\boldsymbol{\Theta}^2}(\boldsymbol{\Theta}^\star + \tau(T(\boldsymbol{\Theta}) - \boldsymbol{\Theta}^\star), \boldsymbol{\Phi}^\star + \tau(u_1(\boldsymbol{\Theta}) - \boldsymbol{\Phi}^\star))d\tau,$$

$$\boldsymbol{H}_{12} \;\; = \;\; \int_0^1 \tfrac{\partial^2 f}{\partial\boldsymbol{\Theta}\partial\boldsymbol{\Phi}}(\boldsymbol{\Theta}^\star + \tau(T(\boldsymbol{\Theta}) - \boldsymbol{\Theta}^\star), \boldsymbol{\Phi}^\star + \tau(u_1(\boldsymbol{\Theta}) - \boldsymbol{\Phi}^\star))d\tau,$$

$$\boldsymbol{H}_{22} \;\; = \;\; \int_0^1 \tfrac{\partial^2 f}{\partial\boldsymbol{\Phi}^2}(\boldsymbol{\Theta}^\star + \tau(\boldsymbol{\Theta} - \boldsymbol{\Theta}^\star), \boldsymbol{\Phi}^\star + \tau(u_1(\boldsymbol{\Theta}) - \boldsymbol{\Phi}^\star))d\tau,$$

$$\boldsymbol{H}_{21} \;\; = \;\; \int_0^1 \tfrac{\partial^2 f}{\partial\boldsymbol{\Phi}\partial\boldsymbol{\Theta}}(\boldsymbol{\Theta}^\star + \tau(\boldsymbol{\Theta} - \boldsymbol{\Theta}^\star), \boldsymbol{\Phi}^\star + \tau(u_1(\boldsymbol{\Theta}) - \boldsymbol{\Phi}^\star))d\tau.$$

*Therefore, assuming that the Hessian is strictly positive definite and Lipschitz continuous in a neighborhood of the solution, localizing the current point to the neighborhood, $\boldsymbol{\Theta} \approx \boldsymbol{\Theta}^\star$ and $\boldsymbol{\Phi} \approx \boldsymbol{\Phi}^\star$, we can obtain the contraction property, as in the previous example (see, e.g., Theorem 1.2.5 in Nesterov (2018) for the local analysis of Newton's method).*

### G.3 LINEAR MODELING AND DECOUPLING

In this section, let us study an important example of *linear models*, applicable to both experts and the router. As we will show, in this case and under very mild assumptions we can justify all conditions from the previous section and therefore obtain the global linear convergence for our alternating process.

**Problem Formulation** For simplicity, we consider the case of one client and assume that training and validation datasets are the same, $\boldsymbol{X}^{\text{train}} = \boldsymbol{X}^{\text{valid}}$. However, our observations can be generalized to a more general case of several clients, and different but statistically similar datasets $\boldsymbol{X}^{\text{train}} \sim \boldsymbol{X}^{\text{valid}}$. Hence, we have, $f_1 \equiv f_2 \equiv f$. Note that in this case, our bi-level formulation is also equivalent to joint minimization of $f$ w.r.t. all variables.

We assume that our client has one generalist expert model, that we denote by $\boldsymbol{\theta}^0 \in \mathbb{R}^d$, and $N \geq 0$ specialist experts, that we denote by $\boldsymbol{\theta}^1, \dots \boldsymbol{\theta}^N \in \mathbb{R}^d$. We compose these models together as matrix $\boldsymbol{\Theta} = (\boldsymbol{\theta}^0, \dots, \boldsymbol{\theta}^N)$. In principle, different models can have different expressivity, which we take into account in our modeling by a convex set of constraints: $\boldsymbol{\Theta} \in Q \subseteq \mathbb{R}^{d \times (N+1)}$.

We denote by $\boldsymbol{\phi}^0, \dots, \boldsymbol{\phi}^N \in \mathbb{R}^d$ the parameters of our Router, composed together as matrix $\boldsymbol{\Phi} = (\boldsymbol{\phi}^0, \dots, \boldsymbol{\phi}^N)$, which can also be constrained by a convex set: $\boldsymbol{\Phi} \in \Omega \subseteq \mathbb{R}^{d \times (N+1)}$. For a given data input $\boldsymbol{x} \in \mathbb{R}^d$, the Router decides which experts to use with the SoftMax operation $\boldsymbol{x} \mapsto \boldsymbol{\pi}_{\boldsymbol{\Phi}}(\boldsymbol{x}) \in \Delta_{N+1}$, where

$$
\Delta_{N+1} \quad := \quad \Big\{ \boldsymbol{y} \in \mathbb{R}_+^{N+1} \ : \ \sum_{j=0}^{N} y^{(j)} = 1 \Big\}
$$

is the standard Simplex, and

$$
\pi_{\boldsymbol{\Phi}}^{(j)}(\boldsymbol{x}) \quad := \quad \frac{\exp(\langle \boldsymbol{\phi}^j, \boldsymbol{x} \rangle)}{\sum_{k=0}^{N} \exp(\langle \phi^k, \boldsymbol{x} \rangle)}. \tag{14}
$$

Under these assumptions, we set the following structure of our optimization objective,

$$
f(\boldsymbol{\Theta}, \boldsymbol{\Phi}) \quad = \quad \frac{1}{n} \sum_{i=1}^{n} \ell_i \Big( \sum_{j=0}^{N} \pi_{\boldsymbol{\Phi}}^{(j)}(\boldsymbol{x}_i) \cdot \langle \boldsymbol{\theta}^j, \boldsymbol{x}_i \rangle \Big) + \frac{\alpha}{2} \Big( \|\boldsymbol{\Theta}\|_F^2 + \|\boldsymbol{\Phi}\|_F^2 \Big), \tag{15}
$$

where $\boldsymbol{x}_1, \dots, \boldsymbol{x}_n$ are given data vectors, and $\ell_i(\cdot), 1 \leq i \leq n$ are the corresponding convex losses (e.g. the logistic loss for binary classification, or the quadratic loss for regression problem). We use $\alpha \geq 0$ as a regularization parameter, which can also be seen as the *weight decay*, and $\| \cdot \|_F$ is the Frobenius norm of a matrix.

**Decoupling** Let us introduce *the auxiliary variables*, $\boldsymbol{\lambda}^i \in \Delta_{N+1}$, $1 \leq i \leq n$, and $\boldsymbol{\Lambda} = (\boldsymbol{\lambda}^1, \dots, \boldsymbol{\lambda}^n) \in \Delta_{N+1}^n \subseteq \mathbb{R}^{(N+1) \times n}$, which is a column-stochastic matrix. Employing the matrix notation, we can rewrite our problem in the following form:

$$
\min_{\substack{\boldsymbol{\Theta} \in Q, \boldsymbol{\Phi} \in \Omega \\ \boldsymbol{\Lambda} \in \Delta_{N+1}^n}} \Big\{ \frac{1}{n} \sum_{i=1}^{n} \ell_i \Big( \langle \boldsymbol{\lambda}^i, \boldsymbol{\Theta}^\top \boldsymbol{x}_i \rangle \Big) + \frac{\alpha}{2} \Big( \|\boldsymbol{\Theta}\|_F^2 + \|\boldsymbol{\Phi}\|_F^2 \Big) \ : \ \boldsymbol{\lambda}^i = \boldsymbol{\pi}_{\boldsymbol{\Phi}}(\boldsymbol{x}_i), \ 1 \leq i \leq n \Big\}. \tag{16}
$$

Now, we apply the relaxation of constrained problem (16) by the following *decouple* of $\lambda^i$ from $\boldsymbol{\pi}_{\boldsymbol{\Phi}}(\boldsymbol{x}_i)$, with some parameter $\mu \geq 0$ and a *distance function* $V : \Delta_{N+1} \times \Delta_{N+1} \to \mathbb{R}_+$ between distributions:

$$
\min_{\substack{\boldsymbol{\Theta} \in Q, \boldsymbol{\Phi} \in \Omega \\ \boldsymbol{\Lambda} \in \Delta_{N+1}^n}} \Big\{ F_\mu(\Theta, \boldsymbol{\Phi}, \boldsymbol{\Lambda}) \quad := \quad \frac{1}{n} \sum_{i=1}^{n} \ell_i \Big( \langle \boldsymbol{\lambda}^i, \boldsymbol{\Theta}^\top \boldsymbol{x}_i \rangle \Big) + \frac{\alpha}{2} \Big( \|\boldsymbol{\Theta}\|_F^2 + \|\boldsymbol{\Phi}\|_F^2 \Big)
$$
$$
+ \frac{\mu}{2n} \sum_{i=1}^{n} V(\boldsymbol{\lambda}^i; \boldsymbol{\pi}_{\boldsymbol{\Phi}}(\boldsymbol{x}_i)) \Big\}. \tag{17}
$$

A natural choice for $V$ is the *Kullback–Leibler divergence*, which gives, for every $1 \leq i \leq n$:

$$
\begin{aligned}
V(\boldsymbol{\lambda}^i; \boldsymbol{\pi}_{\boldsymbol{\Phi}}(\boldsymbol{x}_i)) \quad &:= \quad \sum_{j=0}^{N} \big[\boldsymbol{\lambda}^i\big]^{(j)} \ln \big[\boldsymbol{\lambda}^i\big]^{(j)} - \sum_{j=0}^{N} \big[\boldsymbol{\lambda}^i\big]^{(j)} \ln \big[\boldsymbol{\pi}_{\boldsymbol{\Phi}}(\boldsymbol{x}_i)\big]^{(j)} \\
&\overset{(14)}{=} \quad \sum_{j=0}^{N} \big[\boldsymbol{\lambda}^i\big]^{(j)} \Big( \ln \big[\boldsymbol{\lambda}^i\big]^{(j)} - \langle \boldsymbol{\phi}^j, \boldsymbol{x}_i \rangle \Big) + \ln \Big( \sum_{j=0}^{N} \exp \big( \langle \boldsymbol{\phi}^j, \boldsymbol{x}_i \rangle \big) \Big) \\
&= \quad d(\boldsymbol{\lambda}^i) - \langle \boldsymbol{\lambda}^i, \boldsymbol{\Phi}^\top \boldsymbol{x}_i \rangle + s(\boldsymbol{\Phi}^\top \boldsymbol{x}_i),
\end{aligned}
$$

where

$$
d(\boldsymbol{\lambda}) \quad := \quad \sum_{j=0}^{N} \lambda^{(j)} \ln \lambda^{(j)}, \qquad \boldsymbol{\lambda} \in \Delta_{N+1},
$$

is the negative entropy, and

$$
s(\boldsymbol{y}) \quad := \quad \ln \Big( \sum_{j=0}^{N} \exp y^{(j)} \Big), \qquad \boldsymbol{y} \in \mathbb{R}^{N+1}
$$

is the log-sum-exp function. Note that both $d(\cdot)$ and $s(\cdot)$ are convex functions on their domains. Moreover, it is well known that $d(\cdot)$ is *strongly convex* w.r.t. $\ell_1$-norm (see, e.g., Example 2.1.2 in Nesterov (2018)):

$$\langle \nabla^2 d(\boldsymbol{\lambda})\boldsymbol{h}, \boldsymbol{h}\rangle \geq \|\boldsymbol{h}\|_1^2, \qquad \boldsymbol{\lambda} \in \Delta_{N+1}, \boldsymbol{h} \in \mathbb{R}^{N+1}. \tag{18}$$

Therefore, we obtain the following **decoupled optimization formulation**:

$$
\begin{aligned}
\min_{\substack{\boldsymbol{\Theta}\in Q, \boldsymbol{\Phi}\in\Omega \\ \boldsymbol{\Lambda}\in\Delta_{N+1}^n}} \Bigg\{ \; & F_\mu(\boldsymbol{\Theta}, \boldsymbol{\Phi}, \boldsymbol{\Lambda}) \\
&= \frac{1}{n}\sum_{i=1}^{n}\left[\ell_i\Big(\langle\boldsymbol{\lambda}^i, \boldsymbol{\Theta}^\top\boldsymbol{x}_i\rangle\Big) + \mu\Big(d(\boldsymbol{\lambda}^i) + s(\boldsymbol{\Phi}^\top\boldsymbol{x}_i) - \langle\boldsymbol{\lambda}^i, \boldsymbol{\Phi}^\top\boldsymbol{x}_i\rangle\Big)\right] \\
&\quad + \frac{\alpha}{2}\Big(\|\boldsymbol{\Theta}\|_F^2 + \|\boldsymbol{\Phi}\|_F^2\Big) \Bigg\}.
\end{aligned}
\tag{19}
$$

It is clear that setting parameter $\mu := +\infty$, we obtain that (19) is equivalent to our original problem (16). However, for $\mu < +\infty$ we obtain more flexible formulation with auxiliary distributions $\boldsymbol{\lambda}^i \in \Delta_{N+1}$, each for every data sample $1 \leq i \leq n$, that makes it easier to treat the problem. Parameters $(\boldsymbol{\lambda}^i)$ has an interpretation of *latent variables*, which makes our approach similar to the classical EM-algorithm Jordan & Jacobs (1994).

It is clear that function $F_\mu(\boldsymbol{\Theta}, \boldsymbol{\Phi}, \boldsymbol{\Lambda})$ is *partially convex*: it is convex w.r.t $(\boldsymbol{\Theta}, \boldsymbol{\Phi})$ when $\boldsymbol{\Lambda}$ is fixed, and it is also convex w.r.t. $\boldsymbol{\Lambda}$ when $(\boldsymbol{\Theta}, \boldsymbol{\Phi})$ is fixed.

In what follows, we show that under very mild conditions and choosing regularization parameter $\alpha, \mu \geq 0$ sufficiently large, we can ensure that $F_\mu(\cdot)$ is *jointly strongly convex*, regardless of non-convex cross terms: $\ell_i\big(\langle\boldsymbol{\lambda}^i, \boldsymbol{\Theta}^\top\boldsymbol{x}_i\rangle\big)$ and $\langle\boldsymbol{\lambda}^i, \boldsymbol{\Phi}^\top\boldsymbol{x}_i\rangle$. Our theory generalizes a recent approach to soft clustering Nesterov (2020). With this technique, we will be able to show the global linear convergence rate for the alternating minimization approach that we discussed in the previous sections.

**Joint Strong Convexity** Let us consider the $i$-th term of our objective (19) that correspond to the data sample with index $1 \leq i \leq n$. Omitting extra indices, we obtain the following function,

$$F(\boldsymbol{\Theta}, \boldsymbol{\Phi}, \boldsymbol{\lambda}) = \ell\Big(\langle\boldsymbol{\lambda}, \boldsymbol{\Theta}^\top\boldsymbol{x}\rangle\Big) - \mu\langle\boldsymbol{\lambda}, \boldsymbol{\Phi}^\top\boldsymbol{x}\rangle + \frac{\alpha}{2}\Big(\|\boldsymbol{\Theta}\|_F^2 + \|\boldsymbol{\Phi}\|_F^2\Big) + \mu d(\boldsymbol{\lambda}) + \mu s(\boldsymbol{\Phi}^\top\boldsymbol{x}), \tag{20}$$

where $\boldsymbol{\Theta} \in Q$, $\boldsymbol{\Phi} \in \Omega$, $\boldsymbol{\lambda} \in \Delta_{N+1}$. Our goal is to ensure that (20) is strongly convex w.r.t to the standard Euclidean norm of the joint variable. Namely, we establish the following result.

**Proposition 1.** *Let the loss function $\ell(\cdot)$ be convex and assume that its first derivative is bounded: $\rho \geq \max_t \ell'(t)$. Assume that the regularization coefficient is sufficiently large:*

$$\alpha \geq 2\|\boldsymbol{x}\|^2 \max\Big\{\mu, \frac{\rho^2}{\mu}\Big\}. \tag{21}$$

*Then the objective in (20) is strongly convex.*

*Proof.* Note that the log-sum-exp function $s(\cdot)$ is convex. Therefore, it is sufficient to prove that both functions

$$g_1(\boldsymbol{\Theta}, \boldsymbol{\lambda}) := \ell\Big(\langle\boldsymbol{\lambda}, \boldsymbol{\Theta}^\top\boldsymbol{x}\rangle\Big) + \frac{\alpha}{4}\|\boldsymbol{\Theta}\|_F^2 + \frac{\mu}{4}d(\boldsymbol{\lambda}) \quad \text{and}$$

$$g_2(\boldsymbol{\Phi}, \boldsymbol{\lambda}) := \mu\langle\boldsymbol{\lambda}, \boldsymbol{\Phi}^\top\boldsymbol{x}\rangle + \frac{\alpha}{4}\|\boldsymbol{\Theta}\|_F^2 + \frac{\mu}{4}d(\boldsymbol{\lambda})$$

are convex. Computing the second derivative of $g_1$ and applying it to an arbitrary direction $\boldsymbol{z} = [\boldsymbol{H}; \boldsymbol{h}]$ of corresponding shapes, we get

$$
\begin{aligned}
\langle \nabla^2 g_1(\boldsymbol{\Theta}, \boldsymbol{\lambda})\boldsymbol{z}, \boldsymbol{z} \rangle \quad = \quad & \tfrac{\alpha}{2}\|\boldsymbol{H}\|_F^2 + \tfrac{\mu}{4}\langle \nabla^2 d(\boldsymbol{\lambda})\boldsymbol{h}, \boldsymbol{h} \rangle \\
& + \ell''(\langle \boldsymbol{\lambda}, \boldsymbol{\Theta}^\top \boldsymbol{x} \rangle) \cdot \underbrace{\left[ \langle \boldsymbol{h}, \boldsymbol{\Theta}^\top \boldsymbol{x} \rangle^2 + \langle \boldsymbol{\lambda}, \boldsymbol{H}^\top \boldsymbol{x} \rangle^2 + \langle \boldsymbol{h}, \boldsymbol{\Theta}^\top \boldsymbol{x} \rangle \cdot \langle \boldsymbol{\lambda}, \boldsymbol{H}^\top \boldsymbol{x} \rangle \right]}_{\geq 0} \\
& + \ell'(\langle \boldsymbol{\lambda}, \boldsymbol{\Theta}^\top \boldsymbol{x} \rangle) \cdot \langle \boldsymbol{h}, \boldsymbol{H}^\top \boldsymbol{x} \rangle \\
\geq \quad & \tfrac{\alpha}{2}\|\boldsymbol{H}\|_F^2 + \tfrac{\mu}{4}\|\boldsymbol{h}\|_1^2 - \rho\|\boldsymbol{h}\|_1 \cdot \|\boldsymbol{H}\|_F \cdot \|\boldsymbol{x}\| \\
\overset{(*)}{\geq} \quad & \|\boldsymbol{h}\|_1 \cdot \|\boldsymbol{H}\|_F \cdot \left( \sqrt{\tfrac{\alpha\mu}{2}} - \rho\|\boldsymbol{x}\| \right) \overset{(21)}{\geq} \quad 0,
\end{aligned}
$$

where we used Young's inequality in $(*)$. The bound for $g_2$ follows by the same reasoning, substituting $\ell(t) := \mu t$ and therefore setting $\rho := \mu$. $\qquad\square$

For the decoupled optimization formulation (19) it is natural to organize iterations in the following sequential order, starting from an arbitrary $\boldsymbol{\Theta}_0 \in Q$ and $\boldsymbol{\Phi}_0 \in \Omega$, for some $\mu > 0$:

$$
\boxed{
\begin{aligned}
\boldsymbol{\Lambda}_{k+1} \quad &= \quad \underset{\boldsymbol{\Lambda} \in \Delta_{N+1}^n}{\arg\min} \, F_\mu(\boldsymbol{\Theta}_k, \boldsymbol{\Phi}_k, \boldsymbol{\Lambda}), \\
\boldsymbol{\Phi}_{k+1} \quad &= \quad \underset{\boldsymbol{\Phi} \in \Omega}{\arg\min} \, F_\mu(\boldsymbol{\Theta}_k, \boldsymbol{\Phi}, \boldsymbol{\Lambda}_{k+1}), \\
\boldsymbol{\Theta}_{k+1} \quad &= \quad \underset{\boldsymbol{\Theta} \in Q}{\arg\min} \, F_\mu(\boldsymbol{\Theta}, \boldsymbol{\Phi}_{k+1}, \boldsymbol{\Lambda}_{k+1}).
\end{aligned}
}
\tag{22}
$$

Note that each minimization subproblem in (22) is convex and can be implemented very efficiently by means of linear algebra and convex optimization. At the same time, due to decoupling of variables and strong convexity we are able to ensure the global convergence of this process to the solution of (19).

### G.4 CONVERGENCE FOR FUNCTIONAL RESIDUAL

Note that in our decoupled optimization formulation (19), variables $\boldsymbol{\Theta}$ and $\boldsymbol{\Phi}$ are independent of each other, when $\boldsymbol{\Lambda}$ is fixed. Therefore, the second and third step in iteration process (22) can be done independently.

For the sake of notation, let us denote $\boldsymbol{X} \equiv \boldsymbol{\Lambda}$, concatenated variable $\boldsymbol{Y} \equiv (\boldsymbol{\Theta}, \boldsymbol{\Phi})$, and the objective in new variables as $f(\boldsymbol{X}, \boldsymbol{Y}) \equiv F_\mu(\boldsymbol{\Theta}, \boldsymbol{\Phi}, \boldsymbol{\Lambda})$. By our previous analysis, we can assume that $f$ is strongly convex. We denote by $\mu$ the parameter of strong convexity and by $L$ the constant of Lipschitz continuity of the gradient of $f$. Its global minimum is denoted by $(\boldsymbol{X}^\star, \boldsymbol{Y}^\star)$, and correspondingly $f^\star := f(\boldsymbol{X}^\star, \boldsymbol{Y}^\star)$.

Then, iteration process (22) can be rewritten simply as the following alternating iterations, for $k \geq 0$:

$$
\boldsymbol{X}_{k+1} \quad = \quad \underset{\boldsymbol{X} \in \mathcal{X}}{\arg\min} \, f(\boldsymbol{X}, \boldsymbol{Y}_k),
$$

$$
\boldsymbol{Y}_{k+1} \quad = \quad \underset{\boldsymbol{Y} \in \mathcal{Y}}{\arg\min} \, f(\boldsymbol{X}_{k+1}, \boldsymbol{Y}),
$$

where $\mathcal{X}$ and $\mathcal{Y}$ are the corresponding convex domains ($\mathcal{X} \equiv \Delta_{N+1}^n$ and $\mathcal{Y} \equiv \Omega \times Q$).

Then, the stationary condition for $\boldsymbol{Y}_{k+1}$ (see, e.g., Theorem 3.1.23 in Nesterov (2018)) gives

$$
\langle \tfrac{\partial f}{\partial \boldsymbol{Y}}(\boldsymbol{X}_k, \boldsymbol{Y}_{k+1}), \boldsymbol{Y} - \boldsymbol{Y}_{k+1} \rangle \quad \geq \quad 0, \qquad \forall \boldsymbol{Y} \in \mathcal{Y}.
\tag{23}
$$

Choosing

$$
\gamma \quad := \quad \tfrac{\mu}{L} \quad \leq \quad 1,
\tag{24}
$$

we obtain

$$\gamma f(\boldsymbol{X}^\star, \boldsymbol{Y}^\star) + (1-\gamma) f(\boldsymbol{X}_k, \boldsymbol{Y}_{k+1})$$

$$\overset{(*)}{\geq} \quad \gamma \Big[ f(\boldsymbol{X}_k, \boldsymbol{Y}_{k+1}) + \langle \tfrac{\partial f}{\partial \boldsymbol{X}}(\boldsymbol{X}_k, \boldsymbol{Y}_{k+1}), \boldsymbol{X}^\star - \boldsymbol{X}_k \rangle + \langle \tfrac{\partial f}{\partial \boldsymbol{Y}}(\boldsymbol{X}_k, \boldsymbol{Y}_{k+1}), \boldsymbol{Y}^\star - \boldsymbol{Y}_{k+1} \rangle +$$

$$\tfrac{\mu}{2} \|\boldsymbol{X}^\star - \boldsymbol{X}_k\|^2 \Big] \ + \ (1-\gamma) f(\boldsymbol{X}_k, \boldsymbol{Y}_{k+1})$$

$$\overset{(23),(24)}{\geq} \quad f(\boldsymbol{X}_k, \boldsymbol{Y}_{k+1}) + \langle \tfrac{\partial f}{\partial \boldsymbol{X}}(\boldsymbol{X}_k, \boldsymbol{Y}_{k+1}), \gamma(\boldsymbol{X}^\star - \boldsymbol{X}_k) \rangle + \tfrac{L}{2} \|\gamma(\boldsymbol{X}^\star - \boldsymbol{X}_k)\|^2$$

$$\geq \quad \min_{\boldsymbol{X} \in \mathcal{X}} \Big\{ f(\boldsymbol{X}_k, \boldsymbol{Y}_{k+1}) + \langle \tfrac{\partial f}{\partial \boldsymbol{X}}(\boldsymbol{X}_k, \boldsymbol{Y}_{k+1}), \boldsymbol{X} - \boldsymbol{X}_k \rangle + \tfrac{L}{2} \|\boldsymbol{X} - \boldsymbol{X}_k\|^2 \Big\}$$

$$\overset{(**)}{\geq} \quad \min_{\boldsymbol{X} \in \mathcal{X}} \big\{ f(\boldsymbol{X}, \boldsymbol{Y}_{k+1}) \big\} \ = \ f(\boldsymbol{X}_{k+1}, \boldsymbol{Y}_{k+1}),$$

where in $(*)$ we used strong convexity, and in $(**)$ we used the Lipschitz continuity of the gradient. Thus, we get the following inequality:

$$f(\boldsymbol{X}_{k+1}, \boldsymbol{Y}_{k+1}) - f^\star \ \leq \ \big(1-\gamma\big)\Big( f(\boldsymbol{X}_k, \boldsymbol{Y}_{k+1}) - f^\star \Big),$$

and using the same reasoning for $\boldsymbol{Y}_k \mapsto \boldsymbol{Y}_{k+1}$ update, we obtain

$$f(\boldsymbol{X}_{k+1}, \boldsymbol{Y}_{k+1}) - f^\star \ \leq \ \big(1-\gamma\big)^2 \Big( f(\boldsymbol{X}_k, \boldsymbol{Y}_k) - f^\star \Big),$$

which is the global linear rate. Thus, we have established formally the following convergence result.

**Theorem G.2.** *Let $f$ be strongly convex with constant $\mu > 0$, and let its gradient be Lipschitz continuous with constant $L > 0$. Then, for $k \geq 0$ iteration of the alternating minimization process, we have*

$$f(\boldsymbol{X}_k, \boldsymbol{Y}_k) - f^\star \ \leq \ \big(1 - \tfrac{\mu}{L}\big)^{2k} \Big( f(\boldsymbol{X}_0, \boldsymbol{Y}_0) - f^\star \Big).$$

Note that this result is directly applicable for our linear models from previous sections, $f(\boldsymbol{X}, \boldsymbol{Y}) \equiv F_\mu(\boldsymbol{\Theta}, \boldsymbol{\Phi}, \boldsymbol{\Lambda})$, as we show that objective (19) is jointly strongly convex, when the regularization parameter is sufficiently large.