# OpenReview forum: "On-Device Collaborative Language Modeling via a Mixture of Generalists and Specialists"
_ICLR.cc/2025/Workshop/MCDC — MCDC @ ICLR 2025_

### Official Review · Reviewer_q78M · 2025-02-26

**Rating:** 6
**Confidence:** 4
**Fit:** 5

**Summary:**

This paper propose CoMiGS, a modular federated learning architecture for LLM adaptation via a mixture of generalist and specialist LoRA experts. Specifically, CoMiGS is trained using a bi-level optimization objective, alternating between routing and expert parameter optimization. Experiments on GPT-125m and Llama-3.2-1B show superior performance to selected local and federated baselines, while also providing extra insights about the behavior of the framework, theoretically and empirically.

**Reason For Giving A Higher Score:**

Lots of analysis wrt behaviour of model and interaction of router and experts, method seems promising.

**Reason For Giving A Lower Score:**

On-device and federated setup misaligned with actual evaluation.

**Strengths And Weaknesses:**

### Strengths

* The approach of having two sets of experts that focus on global vs. local objectives seems well motivated and turns out to work well, under proper routing.
* I like how the authors showcase the effectiveness of their method compared to baselines.
* I also appreciate the in and out of distribution analyses.

### Weaknesses

* The paper title is somewhat misleading, as it quotes "on-device", but no evaluation has been done there. Moreover, a batch size of 64 might be prohibitive in the memory of edge devices.
* The federated paradigm put forward does not seem to be focusing on cross-device setup, but rather to assume small client sizes and full participation.
* The selection of certain hyperparameters is not properly explained ($\tau$ for router optimization, calibration set extraction)

**Suggestions:**

* It would be insightful to the reader if the authors could provide additional details on the size and extraction method of the validation for reproducibility.
* It would also be interesting to see the behaviour of CoMiGS with other PEFT methods or adapters (e.g. DORA, VERA).
* Another interesting avenue for exploration, especially for on-device deployment, would be the interplay of the technique with quantization methods, where the router and adapters may operate on a lossy pretrained model.
* Since the paper inherits a federated setup, an interesting question arise wrt the tradeoff of utility and privacy when training the generalists under DP.
* $\theta^G$ and $\theta^S$ have not been formally defined as the global and local LoRA parameters.
* The assumed federated setup should be part of the main paper.
* Baselines could be referenced against their original papers in §3.1.
* It is very unclear from the paper how the authors have federated the datasets and whether the distribution is non-IID amongst clients.
* Figures 3-5 font sizes are very small to read.
* Figures 4-5 should have their x-axis label annotated.
* What does the expert average score represent in Figure 4?
* Table 1 should signify what the number in the parentheses signify.
* Does Figure 3 suggest that we might not need the same number of specialists across layers? If so, is there the potential for further optimization?
* The modification of the load-balancing loss with importance weighting could be mentioned from the main text.

---

### Official Review · Reviewer_HJ9M · 2025-02-27

**Rating:** 7
**Confidence:** 3
**Fit:** 4

**Summary:**

The proposed method is a federated/collaborative learning approach where each client is learning a mixture of experts. During a single round, each client locally performs an alternating optimization on its experts and router. Between rounds, a set of generalist experts are sent to a central server for aggregation.

**Reason For Giving A Higher Score:**

The method is well justified, and the authors provide lots of analysis and supporting experiments.

**Reason For Giving A Lower Score:**

The results are a bit dense, and it seems that a significant amount of improvement relies on layer-level routing.

**Strengths And Weaknesses:**

The method appears to perform well -- overall the paper shows that using a mixture of generalist and specialist experts can achieve the benefits of both.

My general criticism is that it is not clear how the generalist experts are determined, nor what the difference is in the generalist vs. expert initialization. During local training, how do you ensure that the experts you have defined as the generalist / expert are indeed learning general / specialized knowledge?

Furthermore, is there a specific reason why the router is not aggregated across all clients? My intuition is that the router should also be a general structure. Like the previous question, it is not clear if better results are simply coming from one particularly strong expert or both experts being universally strong.

The efficiency limitations of applying MoEs should also be clarified. Does FedAvg use 1 or 2 experts? How much benefit is there from using 2 experts over 1?

**Suggestions:**

While the comparison of FedAvg vs 2G or Local vs 2S shows the effectiveness of using a layer-wise router, this contribution does not seems to be the most important part of the paper. The other ablation I reccomend is to use a vanilla router and apply the selective aggregation method e.g. FedAvg-1G1S. This would be similar in spirit to applying partial model personalziation (https://proceedings.mlr.press/v162/pillutla22a.html) to MoEs.

---

### Decision · Program_Chairs · 2025-03-06

**Decision:**

Accept

**Comment:**

This work proposes a mixture of generalist and specialist models, which is very relevant to this workshop. All reviewers recommend acceptance, and we're pleased to accept it to this workshop.